# Intravital imaging strategy FlyVAB reveals the dependence of *Drosophila* enteroblast differentiation on the local physiology

Ruizhi Tang [1,3], Peizhong Qin[1], Xiqiu Liu[2], Song Wu[1], Ruining Yao[1], Guangjun Cai[1], Junjun Gao[1], You Wu[1] & Zheng Guo [1✉]

Aging or injury in *Drosophila* intestine promotes intestinal stem cell (ISC) proliferation and enteroblast (EB) differentiation. However, the manner the local physiology couples with dynamic EB differentiation assessed by traditional lineage tracing method is still vague. Therefore, we developed a 3D-printed platform "FlyVAB" for intravital imaging strategy that enables the visualization of the *Drosophila* posterior midgut at a single cell level across the ventral abdomen cuticle. Using ISCs in young and healthy midgut and enteroendocrine cells in age-associated hyperplastic midgut as reference coordinates, we traced ISC-EB-enterocyte lineages with Notch signaling reporter for multiple days. Our results reveal a "differentiation-poised" EB status correlated with slow ISC divisions and a "differentiation-activated" EB status correlated with ISC hyperplasia and rapid EB to enterocyte differentiation. Our FlyVAB imaging strategy opens the door to long-time intravital imaging of intestinal epithelium.

[1] School of Basic Medicine, Tongji Medical College, Huazhong University of Science and Technology, Wuhan 430030, P.R. China. [2] School of Pharmacy, Tongji Medical College, Huazhong University of Science and Technology, Wuhan 430030, P.R. China. [3]Present address: Department of Medical Laboratory, the Central Hospital of Wuhan, Tongji Medical College, Huazhong University of Science and Technology, Wuhan 430014, P.R. China. ✉email: guozheng@hust.edu.cn

The adult *Drosophila* midgut has been used for decades as a genetic model for studying human intestinal diseases due to its similarities with the mammalian intestine[1–5]. The homeostasis of the epithelium of adult Drosophila midgut is maintained by intestinal stem cells (ISCs), which give rise to either absorptive enterocytes (ECs) or hormone-producing enteroendocrine cells (EEs)[3,6–9]. Each ISC asymmetrically divides under most circumstances, generating one self-renew ISC and one EE progenitor cell (EEP)[3,6] or one enteroblasts (EB)[10]. Most EEPs undergo cell division to yield a pair of EEs, and a few EEPs directly differentiate into a single EE[6]. ISCs express the Notch signaling ligand Delta, which activates the Notch signaling in the EBs to promote their differentiation into ECs[9,11]. Bacteria, chemical injury, or tissue aging causes the turnover of ECs[12–14]. Dying ECs secrete JAK-STAT and EGFR signaling ligands, which evoke the activation of the corresponding signaling pathways in the nearby ISCs and EBs, accelerating ISC proliferation and promoting EB differentiation[15–17]. It is hypothesized that tissue homeostasis is achieved by a balance between the loss of differentiated cells and cell loss inducing daughter cell production and differentiation[18]. However, direct evidence showing this tissue dynamics at a single cell level is still missing.

Genetic lineage tracing is applied to track stem cell lineages in fixing tissues[8,9,16,19,20]. ISCs and their progenies are analyzed in the intestine of flies sacrificed days after lineage induction. However, because lineage analysis is based on a single image taken at the end of a time window in days, the evaluation of the stem cell division rate, the order of progeny generation, and the dynamics of EB differentiation is difficult. Time-lapse imaging is the direct way to resolve the limits mentioned above. A generally used strategy is to dissect the fly intestine in a culture medium and mount it on a dish, providing an ex vivo setup for live imaging in a time window of up to 90 min[21–26]. Beyond the short imaging times, it is still a doubt whether this ex vivo setup can reflect the complexity of the in vivo situation of the ISC lineage. A strategy called "fly mount" extends the time window for time-lapse imaging up to 16 h by pulling out the intestine through a cut in the dorsal abdominal cuticle[27], representing a big step towards the intravital intestine imaging. Recently, "Bellymount" has been proposed for imaging the midgut by lateral compression of the abdomen[28]. The adult *Drosophila* is glued to cover glass, and then, this device enables the intravital tracing of stem cells lineage with intervals of 2–4 days. However, since this strategy uses glue as an intermedium and lateral compression is complicated to restore the gut position, "Bellymount" has its limits in tracking every cell division in ISC lineage over days or revealing the dynamics of EB differentiation.

Here we present a 3D-printed platform "FlyVAB" for the intravital tracing of the lineage in the adult *Drosophila* midgut across the ventral abdomen (VAB) cuticle that becomes transparent thanks to the hyaluronic acid (HA) as the mounting medium. After the vertical compression of the abdomen, we stably recorded the same region of the posterior midgut across the transparent VAB for up to 10 days. The ISC-EB-EC lineage in the posterior midgut can be readily identified using fluorescent markers. Hence, we introduced a Gal4 independent labeling system called the QF-QUAS system, to make the EEs detectable simultaneously with the ISC-EBs. We used the distinctive pattern of ISCs in young and healthy midguts and EEs in age-associated hyperplastic midgut as reference coordinates. We tracked individual ISC lineages over days to study the ISC division rate and EB differentiation. Our intravital imaging strategy reveals a differentiation-poised EB status and a differentiation-activated EB status that was closely correlated with the local intestinal physiology conditions.

## Results

**HA-mediated ventral abdominal transparency and design of the FlyVAB.** We separately tested $H_2O$, 10 mg/ml HA, 5% glycerol, and glue to evaluate which one makes the ventral abdomen of *Drosophila* transparent. All of them enabled the single-cell resolution of the *Drosophila* adult midgut (Supplementary Fig. 1). However, flies were more readily released from $H_2O$ and HA compared with glycerol and glue after intravital imaging. Nevertheless, water rapidly evaporated, while 10 mg/ml HA mimic a glue solution, allowing the placement of the fly ventral abdomen towards to the coverslip; thus, we chose HA rather than $H_2O$ for further experiments. Since flies tended to be stuck on the food due to residual HA on their wings, we removed the wings before the beginning of the experiments of long-term intravital imaging.

We developed the "FlyVAB" using a 3D printing technology for in vivo tracking of individual ISCs and their progenitor cells in the adult *Drosophila* midgut over a long timescale. This platform consisted of three parts: (1) a squeezing system: FlyVAB (P1), (2) an anesthesia system: FlyVAB (P2) and (3) a linker between P1 and P2: FlyVAB (P3). The main features of FlyVAB (P1) were represented by a U-shaped base (Base-1) and a removable lid (Fig. 1a, b; Supplementary movie 1). The inner space of the U-shape was designed to contain a 20× objective. The Base-1 contained a groove for holding a coverslip. A U-shaped water-repellent line was present on the edge of the coverslip, in which 2 µl HA were added to obtain the optical transparency of the cuticle of the ventral abdomen (Fig. 1b; Supplementary movie 1).

To minimize the stress of the flies during compression, a groove was also present at the bottom of the lid leaving a space (5 mm × 10 mm × 0.2 mm) after closure (Fig. 1a). The lid was connected to the Base-1 by a joint-like component to allow its movement forward and backward on the coverslip (Supplementary movie 1). After placing the ventral abdomen of the fly on the HA solution, the vertical pressure exerted on the lid would cause a forward compression in the abdomen of most of the midgut, placing it outside the HA present in the coverslip. Besides, the fly's legs would be stretched under the coverslip, blocking the field of view of the objective lens. Therefore, we applied a two-step process to perform the abdomen squeezing (Fig. 1c, d; Supplementary movie 1). First, the lid was gently pressed to the abdomen. Next, the lid was pulled backward during its moving down. The fly's legs were automatically moved towards the front of the fly during these two compression steps, avoiding blocking the lens.

We designed the FlyVAB (P2) as a $CO_2$ anesthesia system to minimize the motility of the flies (Fig. 1e; Supplementary Fig. 2a). We designed the FlyVAB (P3) to connect FlyVAB (P1) and FlyVAB (P2) to fit into the microscope, and FlyVAB (P3) consisted of a base (Base-2) and a removable stick (S3) (Fig. 1f, g; Supplementary Fig. 2b,c; Supplementary movie 1). We designed a 67.5 degree punching hole in the S3 for holding and adjusting the end of the tube where $CO_2$ passes to anesthetize the flies (Fig. 1f, g).

The key feature of our FlyVAB strategy was that it allowed the easy removal of the flies from the device after the image acquisition, thus allowing their recovery (Supplementary movie 1). After a single squeezing and imaging, flies were able to crawl as usual after their recovery from anesthesia (Supplementary movie 1). Next, we used the manual feeding (MAFE) assay[29] to measure the feeding and digesting of flies after experiencing one time squeezing and imaging in the FlyVAB in order to evaluate the effect of these actions on fly physiology. Both feeding time and food consumption were not significantly different from those in the control group, which was represented

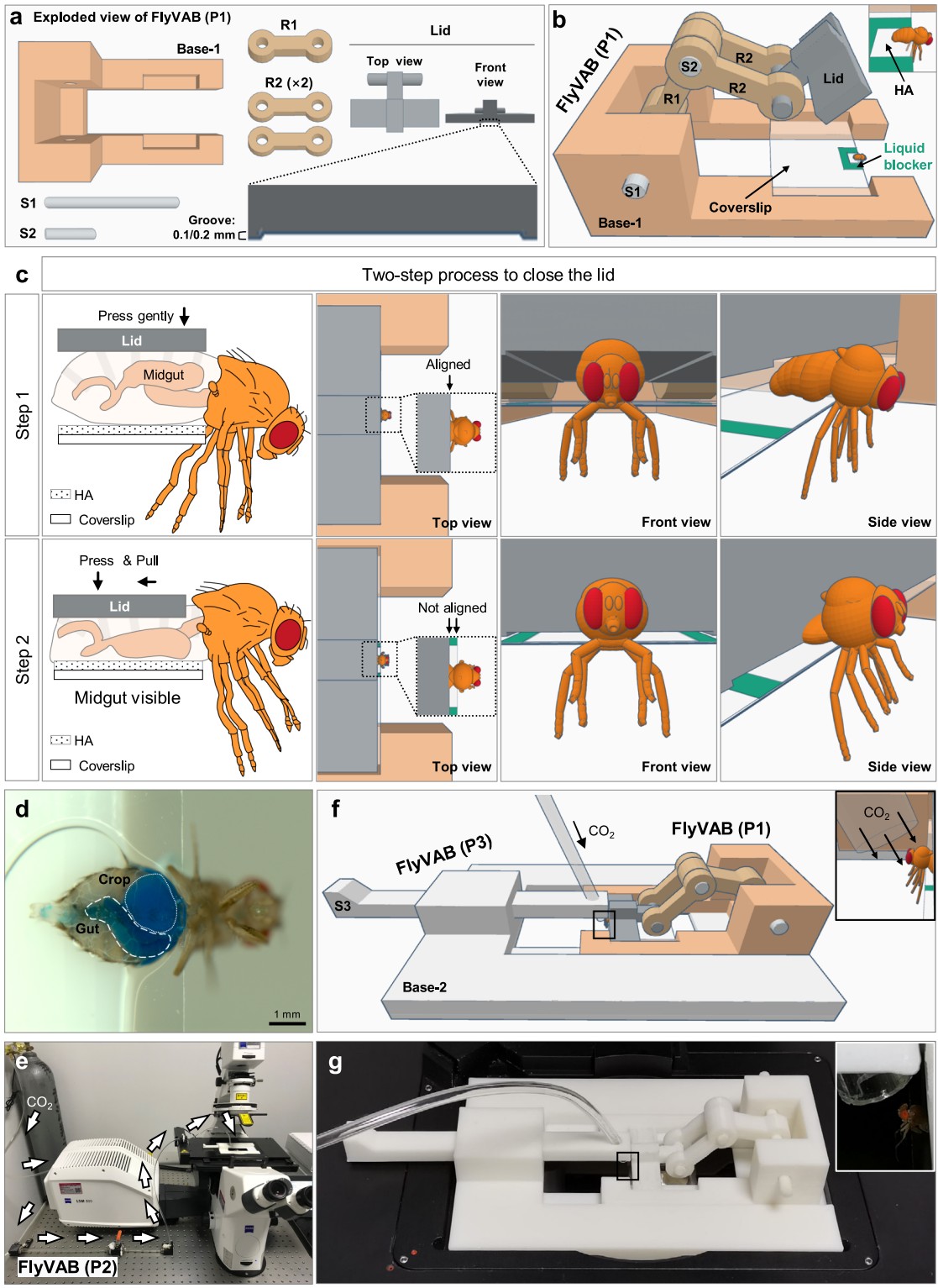

by not-treated flies and with their own wings (Fig. 2a, b). In addition, the flies that were subjected to one-time squeezing had a similar amount of deposits compared with that in the control group at 12 h (Fig. 2c; Supplementary movie 2), indicating that the digesting function of the midgut was not affected after FlyVAB treatment. Furthermore, we calculated the survival curve of 60 control flies and 60 flies subjected to FlyVAB for 7 times at an interval of 2 days (Fig. 2d). Although we removed the wings, flies subjected to squeezing showed a small but significant difference in the number of survived flies compared to the control

group ($P = 0.03$): indeed 97% of flies were alive after loading and releasing every 2 days until day 14. Together, our results suggest that the FlyVAB intravital imaging strategy did not affect flies' digestion and had a small adverse physiologic effect on flies' survival, indicating that the images of the flies acquired from the FlyVAB device represent the in vivo conditions.

**Intravital imaging of ISC lineage in the posterior midgut up to 10 days**. A complete ISC lineage includes ISC, EB, EC, and EE

**Fig. 1 Design of the FlyVAB. a** Exploded view drawing of the FlyVAB (P1). FlyVAB (P1) was designed for imaging the adult *Drosophila* midgut across the ventral abdomen (VAB). It consists of the Base-1, the Lid, one R1, two R2, one S1, and one S2. The lid contains a groove of a height of 0.1/0.2 mm. **b** View of the assembled FlyVAB (P1). FlyVAB (P1) holds a coverslip in the Base-1. A U-shaped water-repelling pattern (green) was drawn on the coverslip to add one drop of HA solution. The adult *Drosophila* was placed into the platform after the removal of the wings. The squared region in **b** is enlarged in the upper right corner. **c** The two-step process to close the lid. First, the lid was aligned to the front edge of the coverslip, then vertically pressed until gently touching the dorsal abdomen cuticle. Second, the lid was pulled backward to the posterior part of the fly while the lid was pressed down until touching the coverslip. **d** Ventral view of the abdomen after closing the lid. The crop and midgut filled with food-stained with a blue dye were both visible through the transparent abdomen. **e** FlyVAB (P2) placed on an optical table, designed as a $CO_2$ anesthesia system. See also Supplementary Fig. 2. **f** View of the assembled FlyVAB (P1) and FlyVAB (P3) together. FlyVAB (P3) was designed as a link between FlyVAB (P1) and FlyVAB (P2) and it consists of the Base-2 and a removable S3. The PU tubing from the FlyVAB (P2) was placed above the fly through the S3 to deliver $CO_2$. The squared region in **f** is enlarged in the upper right corner. See also Supplementary Fig. 2. **g** Image of FlyVAB (P1) and (P3) placed on the microscope stage. The squared region in **g** is enlarged in the upper right corner.

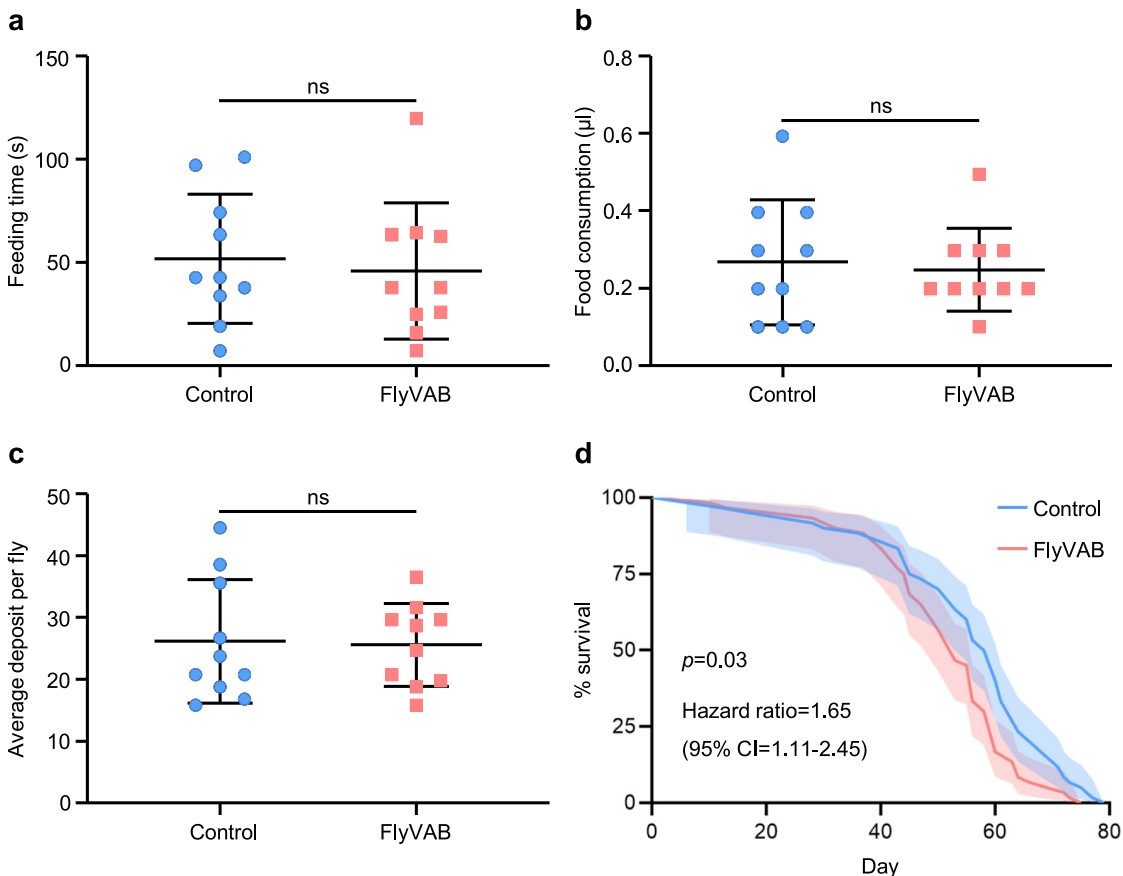

**Fig. 2 Undetectable adverse physiological effects in flies subjected to FlyVAB. a, b** MAFE assay to measure food ingestion after a single compression. The feeding time (**a**) and food consumption (**b**) were quantified in each fly ($n = 10$ per group). Results are shown as mean ± SD. **c** The defecation assay after a single compression. Thirty flies per group were fed with 5% sucrose containing a blue dye, then placed into 10 separated vials. The blue deposit was counted on each vial after 12 h. The average deposit per fly was calculated for each vial. Results are shown as mean ± SD. **d** The survival curve after multiple compressions. A total of 97% flies were alive after being released after loading every 2 days from day 0 to day 14 ($n = 60$ per group), $p = 0.03$ (HR = 1.65; 95% CI = 1.11–2.45). The *Drosophila* strain used for all the experiments was the *Oregon R* (*wild type*). ns, not significant. See the Methods section for all the experimental details.

cells[14,30,31] (Fig. 3a). We used the *esg-Gal4*[8] driving *UAS-tdTomato* to label ISC and EB pairs in the *Drosophila* midgut. Since ISC activates Notch signaling in the EBs, we used the Notch responsive element promoter driving GFP (*NRE-GFP*)[32] to distinguish EBs from *esg*+ ISC-EB pairs (Supplementary Fig. 3a). We could label the EEs using *pros-Gal4*[33] and *Rab3-Gal4*[34] driving GFP. However, to achieve simultaneous but distinct labeling of ISC-EB and EE cells, we generated a *pros-QF* stock by the homology assisted CRISPR knock-in (HACK) method[35] to label EEs by driving *QUAS-mcherry* (Supplementary Fig. 3b), which is

independent of the Gal4-UAS system[36–38]. At last, we used a ubiquitously expressed H2B: RFP fusion protein to label the polyploid nucleus to identify the ECs. All the above labeling methods allow an intravital detection either under a laser scanning confocal microscope, or a widefield fluorescence microscope using the FlyVAB (Fig. 3b).

Next, we wondered which region of the *Drosophila* midgut was examined on the FlyVAB. Ten *esg > tdTomato* flies were subjected to photobleaching, followed by dissection and 4′,6-diamidino-2-phenylindole (DAPI) nuclear staining. Regions that lost the

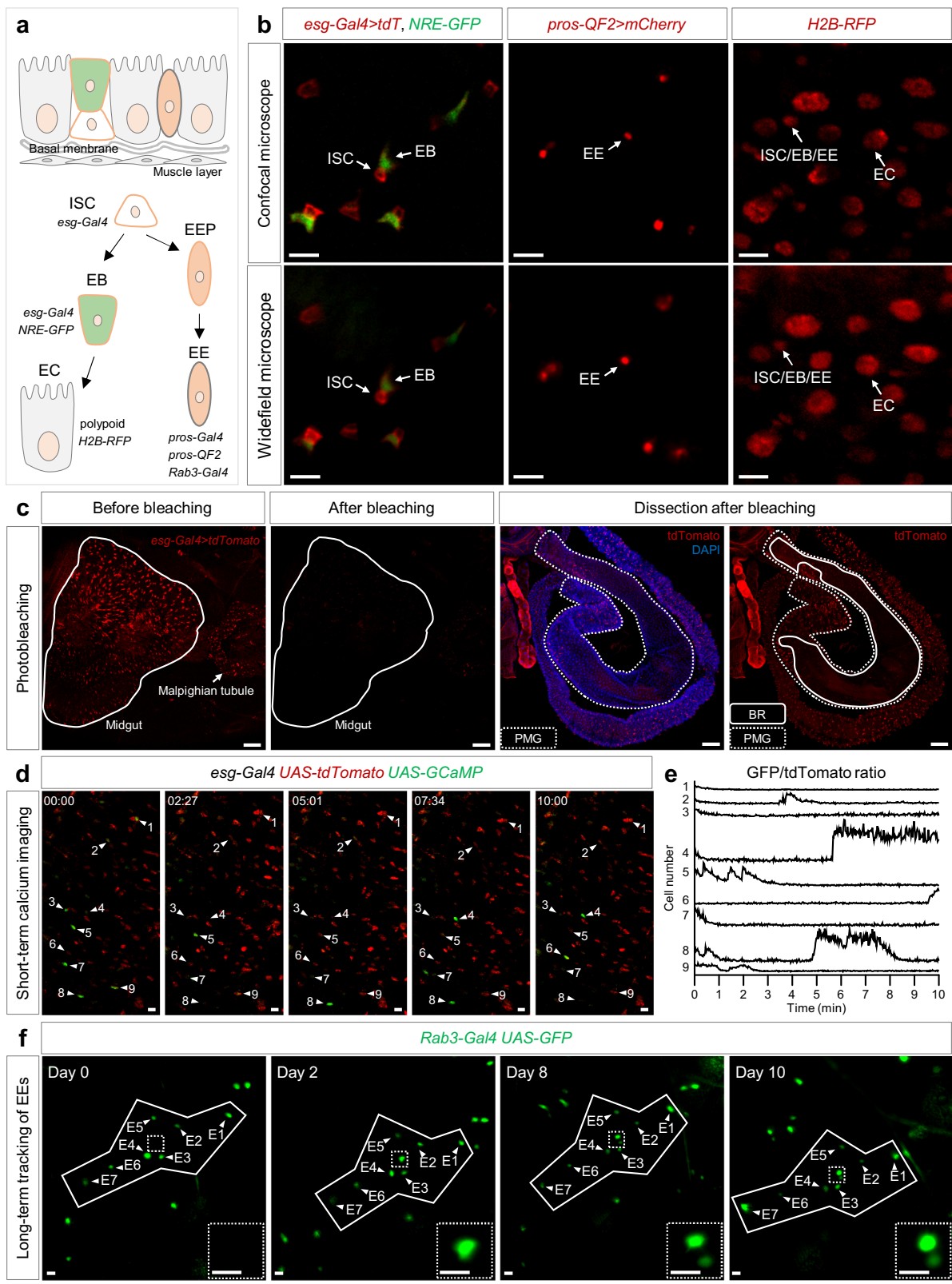

tdTomato fluorescent signal indicated that the FlyVAB setup constantly examined the posterior midgut region 4 (R4)[39] (Fig. 3c; Supplementary Fig. 4).

To validate the usefulness of exploring the rapid change of the in vivo situations by the FlyVAB setup, we applied short-term live imaging of the in vivo ISC calcium oscillation using the FlyVAB device. We used 5 days old female *esg > UAS-mcherry, UAS-GCaMP* flies. ISCs showed an average oscillation rhythm of 1 cycle/10 min (Fig. 3d, e and Supplementary movie 3), demonstrating the potential of FlyVAB to examine quick biological dynamics in vivo.

The next question was whether the same region of the intestine could be re-identified after an interval of hours/days. Since EEs are rarely produced in adult midgut[6], we used the FlyVAB to

**Fig. 3 Intravital short-term live imaging and long-term cell tracing of the posterior midgut. a** Schematic illustration of the adult *Drosophila* midgut. The epithelium mainly consists of four types of cells: intestinal stem cells (ISCs), enteroendocrine cells (EEs), enteroblasts (EBs), and enterocytes (ECs). ISCs and EBs were both labelled by *esg-Gal4*, and they were distinguished by the presence of NRE-GFP in the EBs. EEs were labelled by *pros-Gal4, pros-QF2,* and *Rab3-Gal4*. ECs were identified by the polyploid nuclei. **b** Visualization of the ISC lineage cells using FlyVAB. ISCs (tdTomato+, NRE-GFP-), EBs (tdTomato+, NRE-GFP+), EEs (mcherry+) and ECs (H2B-RFP, polyploid nuclei) visible on a laser scanning confocal microscope or widefield fluorescence microscope. **c** Determination of the visualized region in the midgut by photobleaching. The midgut was barely visible on the FlyVAB device after photobleaching. After dissection and fixation, the bleached region (BR, solid line) was located in the posterior midgut (PMG, dashed line). Genotype: *esg-Gal4 10×UAS-myr:tdTomato*. See also Supplementary Fig. 4. **d** Time-lapse imaging of calcium signal. **e** A total of 354 time points was acquired in 10 min. See also Supplementary Movie 3. **f** Serial imaging of the same region re-identified by a pattern of 7 EEs over 10 days. A pair of newly formed EEs (the dashed square is enlarged in the lower right corner) was identified. Scale bars are 10 μm in **b**, **d**, and **e**, 100 μm in **c**.

perform a long-term tracking of EEs for the feasibility analysis. EEs in the posterior midgut showed distinctive patterns that could be used as reference coordinates (E1–E7 in Fig. 3f). Thus, the corresponding region could be easily re-identified using these coordinates and we could observe one pair of newly formed EEs in a time scale of 10 days (Fig. 3f).

Taken together, our FlyVAB strategy can be used for short-term, short interval, and intravital imaging experimental design. More importantly, FlyVAB can be also used for long-term, hours/days interval lineage tracing of the intravital posterior midgut in the R4 region.

**FlyVAB revealed a "differentiation-poised" EB status in young-heathy intestines.** ISC and EB pairs represent the most common cell distribution pattern in the intestines of young and healthy fly[40,41]. Notch signaling is activated in EBs and many pairs of Notch negative cells also exist in adult intestines[40,41]. However, the Notch activation dynamics in EBs in hours/days are poorly understood, as well as the real-time division rate of each ISC, and the coupling of the ISC division and the EB differentiation dynamics.

Thus, we used 7 days old, healthy female *esg > tdTomato NRE-GFP* flies to perform the intravital imaging by our FlyVAB device (Fig. 4). The tdTomato localized on the membrane represents the ISCs and EBs, and NRE-GFP in the cytoplasm represents the Notch signaling, which also indicates the EB fate[11]. We continuously performed the intravital imaging for 10 days, with an interval of 1 or 2 days. We identified either 4 continuous-time points of the same intestine region in one fly (Fig. 4a), or 3 continuous-time points of the same intestine region in 2 flies (Fig. 4b, c) using the unique patterns of ISCs as reference coordinates (white squared regions in Fig. 4). Among the 9 analyzed ISC-EB clusters that we followed for 2-4 days, only 3 ISCs mitotically divided once (EB[3], EB[7,8] in Fig. 4a, b), indicating that ISCs kept a slow division rate in young and healthy intestines.

We noticed the change of position of ISC and EB in a time course of 2–4 days. The EB[1] did not connect to the ISC at day 0. However, EB[1] connected to the ISC, and the EB[2] moved to the other side of the ISC at day 2 (Fig. 4a). The connected area between ISC and EB could also change over the course of the days (EB[2], EB[4], EB[6], and EB[7,8] in Fig. 4), suggesting that EBs or ISCs did not remain in a fixed position in the intestine epithelium. Furthermore, we calculated the intensity of GFP in EBs and their background (BG), and we normalized the GFP intensity by the signal-to-noise ratio (EB/BG) to represent the Notch activation in EBs over the time course (Fig. 4d). We observed a slow but steady increase of Notch activation in all the EBs connected to the ISCs (EB[1,2], EB[4], EB[6–8] in Fig. 4a, b, d). On the contrary, Notch activation was significantly decreased in EBs detached from the ISCs (EB[5] and EB[9] in Fig. 4b–d). These results suggested that Notch activation is strongly dependent on the connection

between ISC and EB, and Notch activation is reversible in EBs after their detachment from the ligand sending cells, the ISCs.

Taken together, our intravital lineage tracing method revealed a slow ISC division rate in the intestines of young and healthy flies over a time course of 2-4 days. Since Notch was slowly activated in EBs without EC differentiation during this time course, we termed those EBs as in a "differentiation-poised" status in the intestines of young and healthy flies.

**EEs as reference coordinates to trace the EB to EC differentiation.** Since ISC divides slowly in the midguts of young and healthy flies, it is hard to repeat the detection of a specific ISC pattern to re-identify an intestinal region for over 4 days (Fig. 4). To overcome this obstacle, we used the pattern of adult EEs as coordinates to re-identify the same intestinal region in a series of FlyVAB settings. We generated a fly stock, *esg > tdTomato, NRE-GFP; pros-QF > mCherry*, to achieve this goal. Although cells marked by mCherry (EEs) or tdTomato (ISC-EB) were both visible under 561 nm laser, we could distinguish EEs from ISC-EB pairs by the difference in fluorescence intensity. We used the double exposure of imaging acquisition in the same intestine region (Supplementary Fig. 5): first, low laser power and a low digital gain allowed the visualization of mCherry labelled EEs, but not tdTomato labelled cells. Then, high laser power and high digital gain enable the visualization of the ISC-EB pairs, plus the overexposed EEs. Therefore, we unambiguously identified ISCs, EBs, and EEs by comparing these two photos in the same region (Supplementary Fig. 5).

To apply this strategy in the long-term ISC lineage tracing, we imaged a 7 days old adult fly over 109 h with an interval of ~12 h (Fig. 5a, see the full view and two additional z-sections in Supplementary Fig. 6). We re-identified a unique pattern formed by 5 EEs at 8 time points. An ISC is divided once in the intestine region surrounded by these 5 EEs, generating one ISC and one EB in the first 14 h. The NRE-GFP showed the Notch signal activation slowly raising (14–71 h, Fig. 5b) during the first 57 h of EB formation, and then rapidly raising in 15 h (71 h to 86 h, Fig. 5b). This EB cell begins to differentiate into an EC at 86 h onward, and the Notch activity was decreased in the enlarged EB (95 h) and EC (109 h), suggesting that Notch signaling was no longer activated after the EB-EC transition was determined. Taken together, we recorded the ISC-EB-EC transition from a single ISC in the midgut of a living adult *Drosophila*. In vivo Notch signaling activation is much slower in the newly formed EB cell in the intestines of young and healthy flies than in the previous ex vivo results[27].

**FlyVAB revealed a "differentiation-activated" EB status in the intestine with age-associated hyperplasia.** Accumulating evidence suggests that the turnover of ECs induced by local injury leads to accelerated proliferation of ISCs, and promotes the EB to EC differentiation[15–17]. However, the manner ISC-EB pairs and

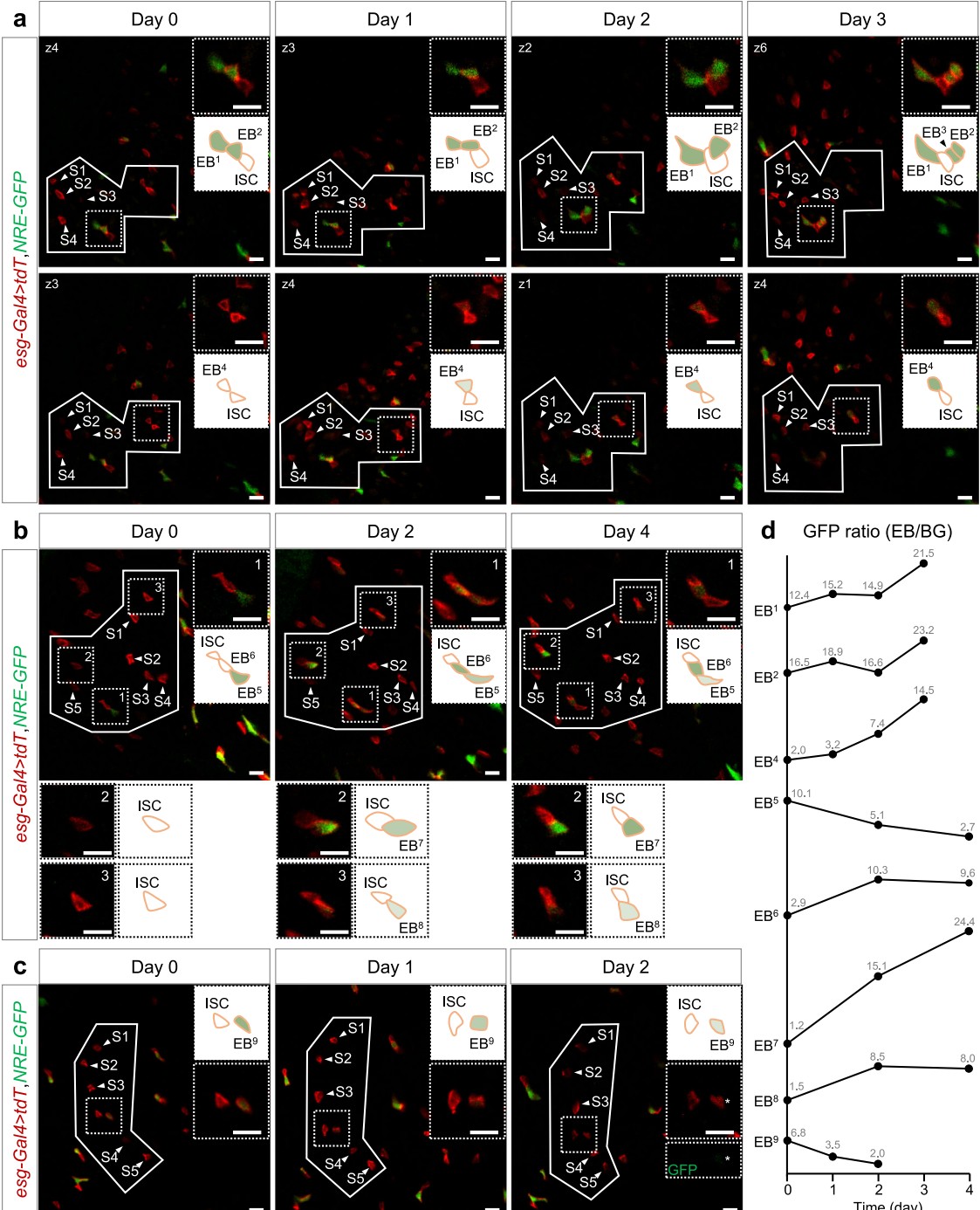

**Fig. 4 Intravital tracking of quiescent ISCs and "differentiation-poised" EBs in the intestines of young and healthy flies. a** Detection of ISCs and EBs over 3 days with the interval of 1 day. A distinctive cell pattern (solid line) formed by four ISCs (S1-S4, white arrowheads) was re-identified over time. An ISC-EB-EB cluster (ISC, EB[1], and EB[2], dashed square in the first panel) and an ISC-EB pair (ISC and EB[4], dashed square in the second panel) were analyzed at two individual z-sections. Newborn EB[3] were identified at day 3. Magnification and schematic illustration inside the dashed squares are shown in the upper right corner. **b** Detection of ISCs and EBs over 4 days with the interval of 2 days. An ISC-EB-EB cluster (ISC, EB[5], and EB[6], dashed box1) and two single ISCs (dashed box2 and box3) were detected using the pattern of ISCs (S1–S5). Newborn EB[7] and EB[8] were identified at day 2. Magnification and schematic illustration inside the dashed squares are displayed in the upper right corner and lower panels. **c** Detection of ISCs and EBs over 2 days with the interval of 1 day. A putative ISC-EB pair (ISC and EB[9], dashed box) was detected using the pattern of ISCs (S1–S5). NRE-GFP in EB[9] (white asterisk) was still visible at day 2. Magnification and schematic illustration inside the dashed squares are shown in the upper right corner. Genotype in all images: *esg-Gal4 10×UAS-myr:tdTomato, NRE-GFP*. All scale bars in **a**–**c** are 10 μm. **d** The quantification of GFP ratio for EBs from **a**–**c** (except for EB[3]). Measurement of the GFP fluorescence intensity of an EB and its surrounding BG, and Notch activity at different time points were normalized using the "signal-to-noise" ratio.

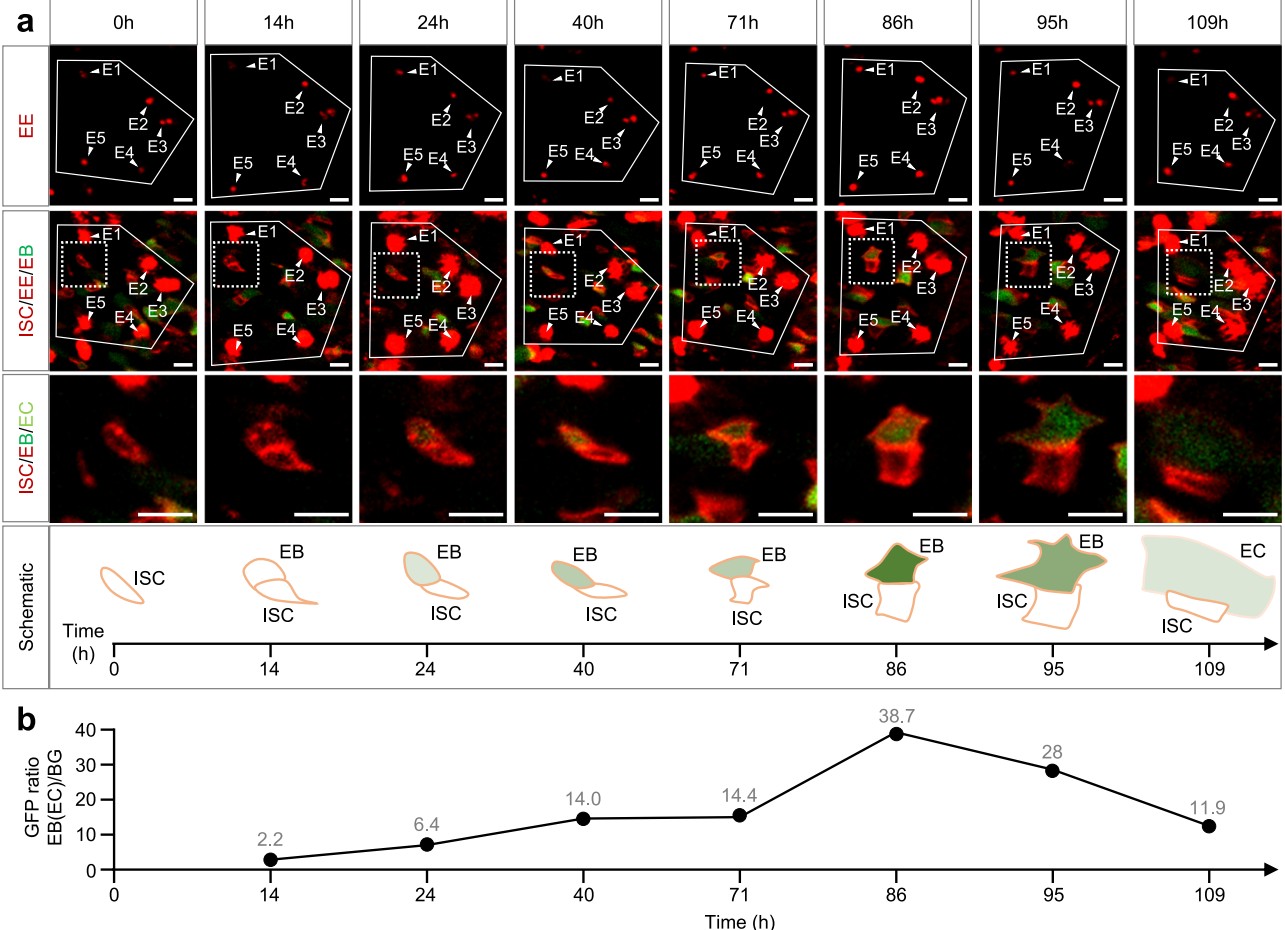

**Fig. 5 Intravital tracking of the ISC-EB-EC transition using the unique pattern of EEs. a** Representative images of the ISC-EB-EC transition. The EE pattern (E1-E5, solid line) was used to trace the same region at 8 time points over 109 h. The EEs were visualized at first by low laser power and low digital gain under the 561 nm laser line (first row). ISCs, EBs, and EEs were subsequently visualized in the same region by higher laser power and digital gain under 488 nm and 561 nm laser line (second row). Magnification and schematic illustration inside the squared area (dashed line) are shown in the third and fourth row. An ISC divided once to generate one ISC and one EB, and this EB further differentiated into an EC. The intensity of the green color indicates the amount of NRE-GFP. Genotype: *esg-Gal4 10×UAS-myr:tdTomato, NRE-GFP; pros-QF2 20×QUAS-6×mcherry*. Scale bars, 10 μm. **b** Quantification of the GFP ratio in the new generated EB (EC) during the ISC-EB-EC transition. The value of the GFP ratio is indicated at each time point.

Notch signaling respond to this differentiation is largely unknown. The intestinal epithelium of aged *Drosophila* has a high probability to lose tissue homeostasis, thus reaching a hyperplastic status[14]. Therefore, using EC turnover marker *upd3-LacZ*[16], midgut hyperplastic marker *vn-LacZ*[15], and ISC mitotic division marker PH3[16], we characterized that 14 days old fly intestines were in a mild hyperplasia state compared with young fly intestines and bleomycin injured[12,42] intestines (Supplementary Fig. 7a–d), which were ideal for long time intravital imaging tracing experiments. Since the intestinal regions could be easily re-identified by our FlyVAB strategy using the unique patterns of EEs, we analyzed the patterns of EEs in the *esg > tdTomato, NRE-GFP; pros-QF > mCherry* intestines from 14 days old flies at 5 time points over 48 h (Fig. 6 and Supplementary Fig. 7e and f).

In one sample, we identified a 3-cell cluster containing 1 ISC and 2 EBs at 0 h by 5 z-sections (two sections are shown in Fig. 6a). EB[1] differentiated into EC[1] at 14 h. In the meantime, Notch activity dramatically increased in EB[2] at 14 h. This EB[2] started to differentiate at 23 h and became an EC[2] at 36 h. During the EB[2] differentiation, ISC divided again, producing an EB[3] at 37 h. Ten hours later, an EB[4] was produced at 47 h by ISC that underwent the third mitotic division. Notably, the EB[3] showed a rapid increase of the Notch signaling to a high level in 10 h

(Fig. 6b). In a second sample, an ISC was divided twice in 47 h and Notch signaling was dramatically activated in the newly formed EB[2] in 24 h (Supplementary Fig. 7e and f). In conclusion, Notch signaling was dramatically activated in EBs in 10–24 h, followed by an EB to EC transition in the intestine with age-associated hyperplasia. At the same time, ISC underwent rapid mitotic divisions associated with the EB to EC transition. This rapid completion of EC differentiation represents a "differentiation-activated" EB status.

## Discussion

In this work, we present a 3D-printed platform called FlyVAB (Fig. 1) used for long-term live imaging of the adult *Drosophila* midgut with a single-cell resolution. This platform is easy to use under an inverted confocal laser scanning microscope or an inverted fluorescence microscope (Fig. 3). We visualized ISCs, EEs, EBs, and newly formed ECs in the same intestine region by vertical compression of the abdomen, using the Gal4/UAS system in combination with the QF/QUAS system (Supplementary Fig. 5). We performed the intravital tracking of ISC proliferation and EB differentiation over days with intervals of 12–48 h using the unique pattern of ISCs or EEs as the reference coordinates

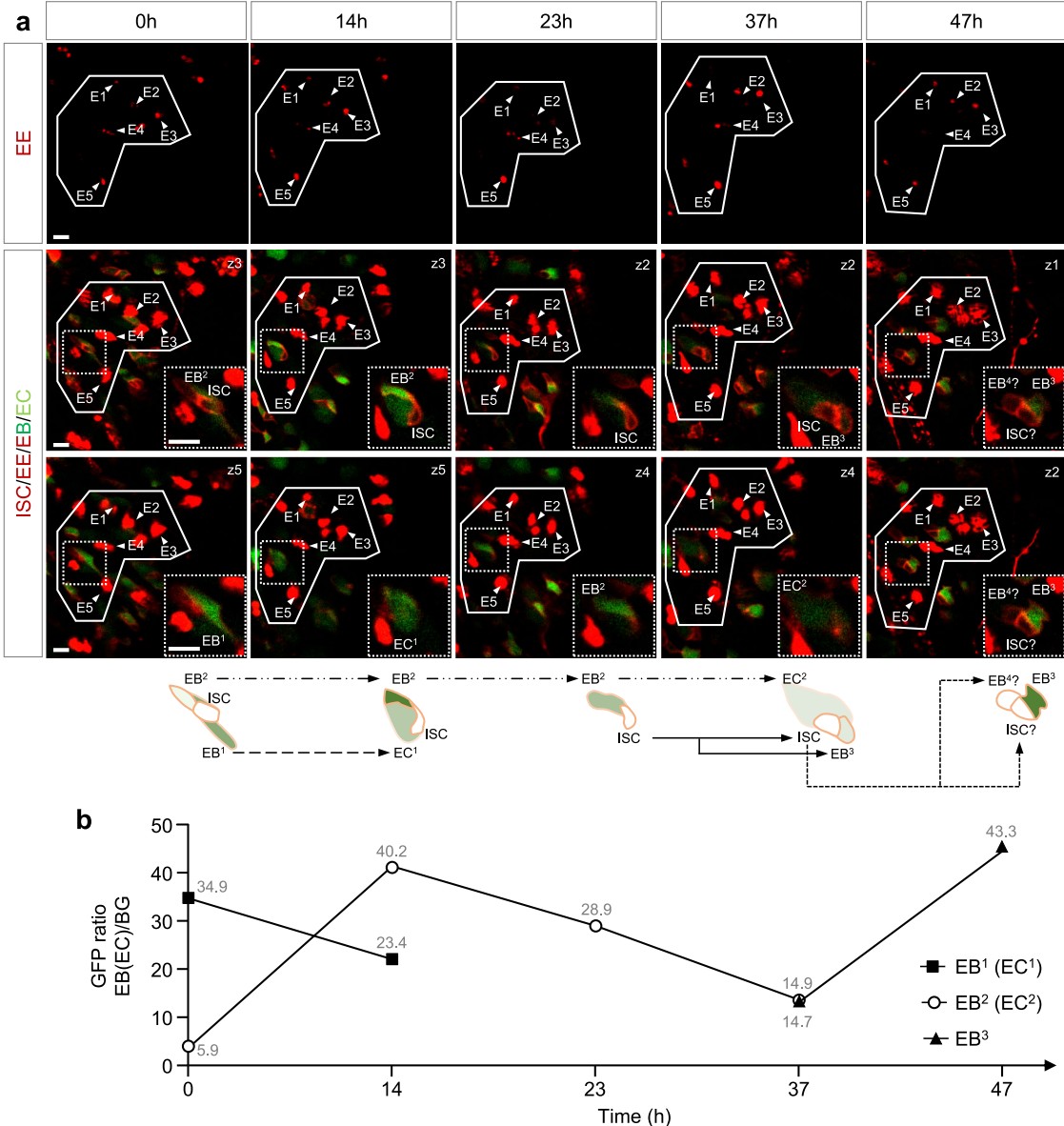

**Fig. 6 A "differentiation-activated" EB status in the intestine with age-associated hyperplasia. a** Representative images of ISC proliferation and EB differentiation in an intestine with age-associated hyperplasia. A pattern of 5 EEs (E1-E5, solid line, first row) was used as a coordinate to re-identify the same intestinal region. An EB-ISC-EB cluster (ISC, EB[1], and EB[2], the dashed square is enlarged in the lower right corner) was tracked at 5 time points over 47 h. Two EB-EC transitions (EB[1] to EC[1], and EB[2] to EC[2]) were identified over 37 h. The ISC was divided once to generate an ISC and an EB[3] along with the EC[2] generation. Shortly afterwards, ISC divided again to generate two cells, which probably were one ISC and one EB[4]. Genotype: *esg-Gal4 10×UAS-myr:tdTomato, NRE-GFP; pros-QF2 20×QUAS-6×mcherry*. Scale bars, 10 μm. **b** Quantification of the GFP ratio in EB[1] (EC[1]), EB[2] (EC[2]), and EB[3]. The value of the GFP ratio is indicated at each time point.

(Figs. 4–6; Supplementary Fig. 7). We observed the ISC-EB-EC transition derived from a single ISC in a living adult *Drosophila* (Fig. 5). Our results suggest a physiological dependence of EB differentiation in the *Drosophila* intestines (Fig. 7): a "differentiation-poised" EB status corresponding to the lack of local damage in the intestines of young and healthy flies, and a "differentiation-activated" EB status corresponding to the local EC damage in the intestine with age-associated hyperplasia.

FlyVAB reveals that the NRE-GFP representing Notch activation was not achieved immediately in the EBs, in agreement with a previous study[27]. The full activation of Notch signaling in newly formed EB could take days in the intestines of young and healthy flies (Fig. 4). In addition, NRE-GFP representing Notch activation could be decreased in EBs when they were no longer

connected with the ISCs (Fig. 4). Since ISCs are cells without Notch, a longer intravital lineage detection in the intestines of young and healthy flies should be considered to explore if those Notch decreasing EBs could dedifferentiate into dividing ISCs, and this approach can represent an intriguing task for future studies.

ISC proliferation rate was investigated using lineage tracing methods in fixed intestines. Ohlstein et al. suggested that the average ISC proliferation rate is one division per day in the posterior midgut[9]. In addition, Liang et al. claimed that ISCs achieve a completed epithelium replacement in 4 days after lineage induction in the hairpin region of the posterior midgut[18]. However, our direct intravital results suggested that the rate of ISC division was so low that it was close to the quiescent status

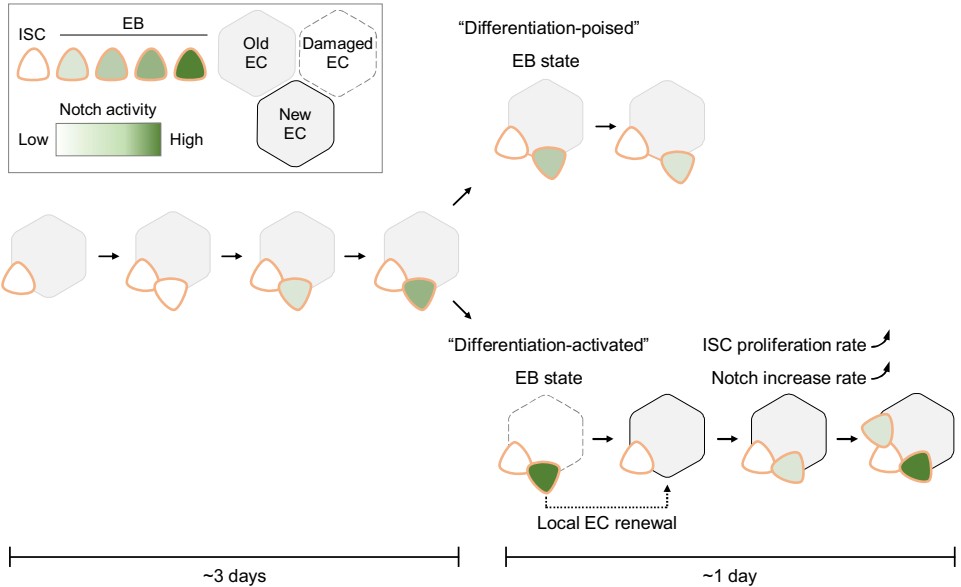

**Fig. 7 Model of physiological dependence of EB differentiation in the Drosophila intestines.** The figures from the left side to the right, show an ISC that is divided once generating one self-renewal ISC and one EB with the activated Notch signaling. Notch signaling gradually increased in this EB in the intestines of young and healthy flies. This process could take more than 3 days. Later on, there were two distinct EB statuses depending on the local physiology: (1) A "differentiation-poised" EB status, corresponding to no EC damage nearby. Notch activation in this "differentiation-poised" EB could be even decreased due to the detachment from the ISC. (2) A "differentiation-activated" EB status, corresponding to a damaged EC nearby. ISC proliferation and Notch activation in EBs were dramatically increased. The local EC renew could be thus achieved in less than one day.

(Fig. 4) in the intestines of young and healthy flies. Nevertheless, the rate of ISC division could reach 2 times in 24 h in the intestine with age-associated hyperplasia (Fig. 6; Supplementary Fig. 7). Also, the high frequency of ISC divisions in aged fly intestines by FlyVAB excluded the possibility that slow division of ISCs were caused by abdominal compression. Therefore, our hypothesis is that there is not any fixed ISC division rate in the intestinal epithelium, and the rate of ISC division depends on the local physiological conditions.

The FlyVAB intravital imaging strategy not only enables the long-term detection of the ISC lineage, but it is also applicable on short intervals time-lapse movies such as $Ca^{2+}$ oscillation and ISC asymmetric divisions. Hence, our FlyVAB strategy can also be used for the intravital observation in other insects with translucent abdomens, such as mosquitoes.

## Methods

**Drosophila stocks and husbandry.** NRE-GFP (BL30727), 20×QUAS-6×mcherry (BL52268), pros-Gal4 (BL84276), UAS-his-RFP (BL56555), 20×UAS-IVS-GCaMP6s (BL42749), and Rab3-YFP (BL62541), vn-lacZ (BL11749) were obtained from the Bloomington Stock Center. pros-QF2 was generated from pros-Gal4 by the homology assisted CRISPR knock-in (HACK) method[35]. esg-Gal4, UAS-mCD8:GFP, Rab3-Gal4, esg-GFP, and ubi-H2B:RFP were kindly donated from the laboratory of Benjamin Ohlstein, while 10×UAS-mry:tdTomato was kindly donated from the laboratory of Ken Irvine. Upd3-LacZ was kindly donated from the laboratory of Bruce A. Edgar.

We performed all Drosophila crosses at 25 °C. The number of females was no more than 30 for each cross to maintain moderate body size, and they were transferred to new bottles containing fresh food every 2 days. Drosophila food was composed of a mix of 12 L water, 0.9 kg cornmeal, 0.12 kg soybean meal, 0.1 kg agar, 0.21 kg yeast, 0.8 l barley malt syrup, and 150 ml 10 % nipagin in ethanol. We collected the females after eclosion and placed them with males in a new vial at 25 °C. We supplied fresh food vials every 2–3 days and we used the 7-day old or 14 days old mated females in the experiments.

**Generation of pros-QF QUAS-mcherry stock.** pros-Gal4: w[1118]; P{w[+mW.hs] =GawB}pros[V1]/TM3, Sb[1] (BL#84276) was crossed with y[1] M{w[+mC] =Act5C-Cas9.P}ZH-2A w[*]; P{3xP3-RFP = QF2.G4HACK}86E11 (BL#66499)[35]. We screened out only one w[+], 3xP3-RFP[+] male fly from over 2000 F2 male flies in

100 F1 crosses. This w[+], 3xP3-RFP[+] fly was confirmed as being a pros-QF by DNA sequencing and QUAS-mcherry driving test.

pros-QF was recombined with 20×QUAS-6×mcherry (BL52268) on the third chromosome to label the EE cells independently of the Gal4-UAS system.

**Fabrication of the FlyVAB.** FlyVAB consisted of an imaging system (P1), an anesthesia system (P2), and a link (P3) between P1 and P2. This platform was mainly manufactured by 3D printing technology. The 3D models were designed by Tinkercad (https://www.tinkercad.com), then converted to an STL file format by the slicing program Creality Slicer (Shenzhen Creality 3D Technology) before loading them into the Ender-3S 3D printer (Shenzhen Creality 3D Technology). All components were printed using polylactic acid (Shenzhen Creality 3D Technology) by the following parameters: layer height 0.1 mm (for P1) or 0.2 mm (for P2 and P3), shell thickness 1.2 mm, bottom/top thickness 1.2 mm, infill 100% (for P1) or 50% (for P2 and P3), and print speed 50 mm/s. FlyVAB (P2) was set up on an optical table by assembling the 3D-printed components, pneumatic fittings, and polyurethane tubing (JEND).

**Preparation of the instrument for intravital imaging of the midgut.** After we placed a cover glass onto Base-1 (Fig. 1b), we drew a U-shaped pattern (forming a square of ~5 mm × ~5 mm × ~5 mm) on its edge using the circle writer (CIRISC), in which we added 2 μl of 10 mg/ml HA (Lifecore, MW 360,000 g/mol). One cover glass could be used for two flies. We anesthetized the flies by $CO_2$, and we removed both wings by a pair of forceps. Next, we roughly placed the abdomen of the fly onto the HA solution and we adjusted its position using the forceps.

As regards the compression of the abdomen, we first moved the lid above the fly and aligned it to the edge of the cover glass. Next, we vertically pressed the lid until it just hit the dorsal abdominal cuticle. While continuing to press the lid to the cover glass, we pulled the lid towards the posterior part of the fly until the legs were no longer lying beneath the cover glass. The compression lasted for 2–5 min for each fly in an image session.

Before imaging, we delivered $CO_2$ through the FlyVAB (P2) to minimize the midgut movement. We first placed FlyVAB (P3) on the stage of the microscope. We then placed FlyVAB (P1) onto FlyVAB (P3) and moved S3 until the PU tubing from FlyVAB (P2) was placed above the fly (Fig. 1f and g).

**Release of flies from FlyVAB.** At the end of the imaging we switched off the release of the $CO_2$ and we opened the lid. Next, we deposited the FlyVAB (P1) on a piece of paper (Kimtech Science, S-200), and we transferred the fly on the paper using a small paintbrush, using the paper to gently roll back and forth the fly to remove the residual HA solution.

**Ingestion assay**. We used the MAFE assay as previously described[29] to quantify the ingested food by the flies. Briefly, the 5-6-day old flies were fasted for 36 h by placing them in vials containing only water. Before fasting, we subjected the flies in the FlyVAB group to compression for ~3 min after the removal of the wings. We individually placed the flies into a 200 μl pipette (one fly per tip), blocked by cotton (10 flies for each group), then the proboscis of the flies was exposed after we carefully removed the pipette tip. Afterwards, 5 μl sucrose 5% containing 0.25% (v/v) blue dye (AmeriColor, Soft Gel Paste 102) were placed in a glass capillary (World Precision Instruments, 1B100F-4) to deliver the food to the proboscis. The feeding event was counted when the fly started drinking the sucrose solution. Once the fly retrieved the proboscis, we repeated the food stimulation until they were unresponsive after 10 cycles. We recorded the start and stop points of each feeding event to calculate the total feeding time. We measured the length of the blue liquid in the capillary before and after the assay by a Vernier caliper (Shanggong) to calculate the food consumption.

**Defecation assay**. We performed the defecation assay according to the method previously described with slight modifications[43]. We first fed the 5–6-day old flies placing them in vials containing 5% sucrose/blue dye for 24 h. Before feeding, we subjected the flies in the FlyVAB group to compression for ~3 min after the removal of the wings. We then divided 30 flies in each group into 10 new vials. The filter papers soaked with 5% sucrose/blue dye were placed at the bottom of each vial. We counted the blue deposits on the vial wall and foam cap after 12 h.

**Survival assay**. We divided 5-day old flies into two groups: control, $n = 60$ and FlyVAB, $n = 60$. We kept 3 flies in each group in a vial containing food at 25 °C and transferred to a vial containing fresh food every 2–3 days. Since flies became easily attached to the food after the removal of the wings, we inserted a piece of filter paper into each vial, keeping all vials in a horizontal position. We loaded the flies in the FlyVAB group into the platform every 2 days from day 0 to day 14. We released them from the FlyVAB after compression for ~3 min each time. We did not perform any compression or removal of wings in the control group. We recorded daily the survival of the flies.

**Microscopy**. We acquired the Brightfield image using a Zeiss Axio Zoom.V16 stereo microscope, and the confocal and wide-field images using an inverted Zeiss LSM800 microscope equipped with a 20× air objective (0.8 NA) and a 10× air objective (0.45 NA). We also acquired the wide-field images using an inverted Zeiss Axio Vert.A1 microscope equipped with a 20× air objective (0.8 NA). As regard serial imaging, we acquired the 10–22 μm Z-stack images (6–12 slices) at 1024 × 1024 pixels with a pixel time of 0.52 μs.

**Calcium imaging**. We used flies carrying *esg-Gal4 10×UAS-mry:tdTomato 20×UAS-IVS-GCaMP6s* for short-term calcium imaging. We loaded the female flies at 6 days after eclosion into the FlyVAB and we mounted them using HA 20 mg/ml. The posterior midgut was imaged using Zeiss Definite Focus. We acquired 6 μm thick Z-stack images (4 slices) at 256 × 256 pixels, with a pixel time of 2.06 μs and a total of 354 time points over 10 min.

**Fly tracking**. We removed the wings of the flies one day prior to imaging, and individually placed the flies in a vial containing food. We placed all vials in a horizontal position as described above. We recorded the shape of the midgut at the first time point, and then, we obtained the images for all the observable regions. At the late time points, after we compared the tissue shape and cell pattern (of quiescent ISCs or EEs) with previous data records, a quick decision had to be made on whether to take images of the gut region or not. We sacrificed the flies when they no longer had re-identifiable tissue regions, and we prepared new flies after each experiment session.

We first exported the images at each time point by ZEN (Zeiss) to label the re-identifiable region and we manually compared them by eyes. Next, we re-examined each image file at each individual z-section, and we chose the most representative image of the labelled region at each time point. The main criterion for selecting representative images is that both of the interested ISCs and EBs are clear in the same focal plane. We finally adjusted the selected images by GIMP (https://www.gimp.org) and Fiji image J (NIH).

**Determination of the visualized region of the midgut**. We performed photobleaching in flies carrying *esg-Gal4 UAS-myr:tdTomato* using an inverted Zeiss LSM800 microscope equipped with a 10× air objective. Briefly, we first visualized most of the abdomen under 561-nm-wavelength laser with a power of 0.5–2% and a zoom of 0.5. We continuously exposed it to 100% laser power until its fluorescence was barely visible. We obtained the images before and after photobleaching using the same laser power. After bleaching, we dissected the tissue in PBS and fixed by 4% formaldehyde (Sigma-Aldrich) for 1 h. After three washes with PBS (5 min × 3), we stained the tissue using DAPI (1 μg/ml, Sigma-Aldrich) for 5 min, followed by three additional washes with PBS. Then we mounted the tissue on a coverslip in which we previously placed 10 μl glycerin 70% and imaged it using a

10 × air objective. We acquired 72–104 μm thick Z-stack images (10–14 slices) at 1024 × 1024 pixels with a zoom of 0.5 and pixel time of 0.76 μs.

**Immunostaining**. We dissected the samples in PBS and fixed in 4% formaldehyde for 3 hours. Primary antibodies were the following and used at the following dilutions: chicken anti-GFP (Abcam, Ab13970, 1/10000 dilution), mouse anti-Pros (Developmental Studies Hybridoma Bank, AB_528440, 1/100 dilution), rabbit anti-phospho-histone H3 (H3ser10, Millipore #2465253, 1/10000 dilution), and chicken anti-β-galactosidase (Abcam, Ab9361, 1/10000 dilution). Samples were incubated with primary antibodies for 3 h at 25 °C or overnight at 4 °C. After three washes with PBT, we incubated the samples with AlexaFluor–conjugated secondary antibody (Thermo Fisher Scientific, dilution at 1/4000) at 25 °C for 3 h. Next, we stained the samples with 1 μg/ml DAPI (Sigma-Aldrich) for 5 min, then we washed them three times with PBT, and we mounted them in 70% glycerin.

**Bleomycin treatment**. We prepared 25 μg/ml bleomycin (Sigma Aldrich) solution with 5% sucrose. Before treatment, flies were starved in empty vials for 3 h. Then we transferred flies into vials containing bleomycin-soaked paper. One day after bleomycin treatment, flies were sacrificed for immunostaining.

**Statistics and Reproducibility**. We performed the statistical analysis by the *t*-test using GraphPad Prism 8 (GraphPad software). As regards the ingestion and defecation assays, we performed the experiments in 10 flies for each group and we presented the results as mean ± SD. As regards the survival assay, we recorded the dead flies in each group every day. We calculated the raw mean values of the GFP of EBs and of three adjacent BGs by Fiji image J to normalize the intensity of the NRE-GFP. We calculated the GFP ratio of EB/BG for each EB at each time point. We generated all graphs by GraphPad Prism 8.

**Reporting summary**. Further information on research design is available in the Nature Research Reporting Summary linked to this article.

## Data availability

The datasets generated and analyzed during the current study are available in the Figshare repository[44]. There is no restriction about data availability.

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

## Acknowledgements

We thank Dr. B. Ohlstein (Columbia University), Dr. Y. Song (Peking University), Dr. Z. Zhai (Hunan Normal University), and Dr. S. Jin (Hubei University) for fly strains. This work was supported by the National Natural Science Foundation of China (31970817 and 31771625 to Z.G., 81803108 to X.L., 31900618 to S.W.), and China Postdoctoral Science Foundation Grant (2017M622408 to S.W.).

## Author contributions

R.T., P.Q., X.L. and Z.G. designed the experiments and wrote the manuscript. R.T. and X.L. performed 3D design and 3D printing. R.T. performed and analyzed experiments. S.W. generated transgenic flies. R.Y. assisted with fly husbandry. G.C. helped to design the $CO_2$ delivery system and assisted with microscopy. J.G. and Y.W. assisted with the ingestion and defecation assay.

## Competing interests

The authors declare no competing interests.
