## [Transparent Peer Review File · Communications Biology]

Reviewers' comments:

Reviewer #2 (Remarks to the Author):

Tang et al. describes a device and technique for live imaging of adult *Drosophila* intestine in situ in a manner that maintains normal intestinal physiology. The authors use this technique to monitor the Notch-mediated differentiation of enteroblast cells from intestinal stem cells over a time window of days, using fluorophore bleaching and labeled enteroendocrine cells to register images from multiple imaging sessions. They identify two different states of enteroblast cells that can differentiate into enterocytes on different time courses, so called "stand-by" and "ready-to-go" enteroblasts, which are states determined by local physiological conditions in the abdomen.

General comments

This paper is basically a description of a device used to hold the abdomen of an adult *Drosophila* for imaging. The biological findings are incidental. The paper would fit well as a technical note in a specialist journal followed by a second publication describing a more in-depth study of stem cell differentiation with information about inter-cellular signaling. Or, this could be a method presented as part of a larger study of ISC differentiation. Right now, it's not enough of a method to justify a methods paper and it isn't enough biology to justify a biology paper. The paper will have very limited appeal to anybody not interested in imaging through the abdominal wall of living flies. It's an incremental step in improving ISC imaging. If it were me, I would delay publishing this paper until I could make it part of a larger study of ISCs, but I'm in a different position than the authors.

Major issues

1. HA not justified or supported - The chief advantages of HA appear to be that it has a refractive index close to that of water and it doesn't evaporate as quickly as water. These conclusions, however, have not been demonstrated. I suggest exploring other mounting media where the RI can be controlled to best match the tissue and the optics and present quantified results. Compare the signal/background for a variety of fluorophores and a variety of media, such as sucrose solutions, iohexol and halocarbon oil. By the way, signal/background is not the same as signal/noise.
2. mCherry labeling of ee cells is unreliable - It doesn't seem reliable to identify ee cells according to the level of expression of red fluorophore. I suggest using a different fluorophore for these important landmarks. If that is not an option, you could tentatively distinguish between the mCherry and tdTomato with spectral unmixing, if your microscope is capable, rather than relying on signal intensity. The safer alternative is to redesign your flies so that all the labels aren't red. Try miRFP703, for example.
3. Signaling in the abdomen - The authors conclude that damaged or aging abdominal tissues provide different signals to ISCs than young, health tissue does. They cite other work that suggests mechanisms where tissue physiology could affect Notch signaling, however, those signaling mechanisms are not addressed in these experiments. Perhaps the authors could design an experiment that explicitly compares aging and damaged tissue. Since you have optical access to the gut, maybe try photo-ablation or optogenetic manipulation of identified cells. The title of the paper mentions local physiology but the paper doesn't manipulate local physiology. That's an opportunity missed.
4. Figure issue - The figures are too heavily annotated to make interpretation possible. Decrease the thickness and brightness of outlines

Other issues

1. Your wording makes it sound like you invented QUAS.
2. Why did you do calcium imaging? Was it just to show that you could? How do the results compare

to images acquired from gut explants or bellymount gut? If your results are similar, it would be another piece of evidence that your method maintains normal physiology. Also, is 0.5Hz fast enough to see actual calcium dynamics in the gut that aren't associated with cell death? I would also suggest jRGECO1b instead of GCaMP6s for imaging through the abdominal wall because of the decreased scattering at the longer wavelengths. Flies are available from Bloomington.

3. Many of the conclusions about cell fate are based on very low sample sizes (n=1). If you are going to make conclusions about the low impact of your method, you will need properly quantified data and appropriate statistical tests.

4. Why didn't you cut the wings off of the control flies in the survival experiment? That's the only way to draw any conclusions about abdominal compression.

Reviewer #3 (Remarks to the Author):

In this manuscript, Guo and colleagues described a new method for intravital imaging in *Drosophila* and applied it to trace ISC-EB-EC lineages in midgut of young and old flies. The authors first designed a 3D-printed platform—"FlyVAB" and used hyaluronic acid as a mounting medium for periodic in vivo tracing over two weeks. Then by using distinctive patterns of ISCs and EEs as reference coordinates, they traced ISC-EB-EC lineage in the R4 region and found that in young and healthy flies, ISCs divide slowly and most EBs are in a "stand-by" status, while in aged flies with ISC hyperplasia, ISCs undergo rapid mitotic divisions and EBs are in a "ready-to-go" status for rapid differentiation into ECs.

The method developed in this study is an alternative to the previously developed "Bellymount" method, but with improved length of tracking time and easiness in operation. The finding of two different status of intestinal progenitors with this new method is also very interesting. Listed below a few minor points for authors to consider.

1. In the method part the authors described that they examined each image file at individual z-section and chose one representative" image of the labeled region, which came out of Fig.3-6 and some supplementary figures. But the authors need to make sure that the chosen images are truly "representative" for analyzing florescent intensity. What does it look like if acquiring the Z-stack images covering most of the thickness of the targeted cells and processing maximum or average projection of these images?

2. With live imaging, the authors observed that ISCs divide much slower than previously-observed and are close to quiescent. Is it possible that some operations applied to the animals during the experimental process, such as the compression applied to the fly abdomen, may have an impact on ISC behavior and thus cause ISC quiescence?

3. Literally it is not obvious about the difference between "stand-by" and "ready-to-go", how about "differentiation-poised" and "differentiation-activated" instead? Also, can EBs with these two status be separated by molecular or morphological markers, such as the levels of Notch activity?

4. Figure 3A schematic : Similar to EC, EE cells are also derived from ISCs via a transient progenitor (EEP) stage.

5. The authors need to check carefully for spelling and formatting errors.

We would like to express our sincere thanks to the reviewers for the constructive and positive comments.

Replies to Reviewer 2:

Reviewer #2 (Remarks to the Author):

Tang et al. describes a device and technique for live imaging of adult *Drosophila* intestine in situ in a manner that maintains normal intestinal physiology. The authors use this technique to monitor the Notch-mediated differentiation of enteroblast cells from intestinal stem cells over a time window of days, using fluorophore bleaching and labeled enteroendocrine cells to register images from multiple imaging sessions. They identify two different states of enteroblast cells that can differentiate into enterocytes on different time courses, so called “stand-by” and “ready-to-go” enteroblasts, which are states determined by local physiological conditions in the abdomen.

General comments

This paper is basically a description of a device used to hold the abdomen of an adult *Drosophila* for imaging. The biological findings are incidental. The paper would fit well as a technical note in a specialist journal followed by a second publication describing a more in-depth study of stem cell differentiation with information about inter-cellular signaling. Or, this could be a method presented as part of a larger study of ISC differentiation. Right now, it’s not enough of a method to justify a methods paper and it isn’t enough biology to justify a biology paper. The paper will have very limited appeal to anybody not interested in imaging through the abdominal wall of living flies.

It’s an incremental step in improving ISC imaging. If it were me, I would delay publishing this paper until I could make it part of a larger study of ISCs, but I’m in a different position than the authors.

Dear reviewer #2,

Thanks for your inspiring suggestions and the critical comments.

We have spent over 3 years, and evolved dozens of versions of designs to make non-invasively tracking an internal, living ISC simple and easy to use. As we explained below, many of your suggestions we have realized and tested before this submission.

The Notch signaling pathway was initially thought to be activated in the daughter of stem cells within a few minutes to promote the differentiation of EB into EC cells [1-3]. Through a cut-window in the abdomen of flies, Lucy O'brien's group found that activation of Notch takes several hours [4]. Our non-invasive intravital imaging found that the activation of Notch varies from hours to dozens of hours depending on the physiology of the midgut. And our imaging data is the first time observed that ISC

division is coupled with the differentiation of daughter cells. There are few examples in the stem cell field discussing the activation speed of Notch. People know Notch affects stem cell differentiation, but our data suggested that the speed of activation matters. We believe the biology we found here is interesting for many stem cell and Notch signaling researchers.

Lucy O'Brien's group published one *Elife* paper 2018[4] and one *Plos Biology* paper 2020[5] about the imaging method of the fly intestines. Probably that's why you thought our method was an incremental step. However, if anyone tried to do gut intravital imaging in multiple days, there are two key difficulties should be resolved: first, since guts are contracting/rolling in the abdomen, the same region has to be re-found constantly among days. Second, intestinal epithelium is the most dynamic cell layer in the *Drosophila*, even the same gut region could be re-found, how could the same ISC be identified among days?

Lucy O'Brien's paper haven't resolved those two difficulties. By a 3D printing platform and a two-step process to close the lid, we could constantly visit the same region of the gut after releasing flies off from the FlyVAB. Combining the QF-QUAS system, we could use the pattern of enteroendocrine cells as coordinates to re-identify the same intestinal stem cell in a series of FlyVAB settings. We overcome those two difficulties, thus we don't agree with the comments that our method is an incremental step.

We totally agree with you that we should publish this method until we could make it part of a larger study of ISC. However, we are in an urgent situation: according to our university's postdoctoral management regulations, the first author of this manuscript cannot stay in the lab if he couldn't have one publication after 3 years of work. Please understand our position.

Major issues

1. HA not justified or supported - The chief advantages of HA appear to be that it has a refractive index close to that of water and it doesn't evaporate as quickly as water. These conclusions, however, have not been demonstrated. I suggest exploring other mounting media where the RI can be controlled to best match the tissue and the optics and present quantified results. Compare the signal/background for a variety of fluorophores and a variety of media, such as sucrose solutions, iohexol and halocarbon oil. By the way, signal/background is not the same as signal/noise.

Answer: Thank you for your suggestions. First of all, I should make it clear that we use air objectives in the FlyVAB setup for intravital imaging. When the fly abdomen is pressed against the cover glass, the gap between the abdomen cuticle and the cover glass produces light scattering, making the light unable to pass through the cuticle for imaging. H₂O or 10% HA solution we used fills these gaps, thereby making the cuticle

of the fly abdomen pressed against the cover glass transparent (Figure 1C). Therefore, the refractive index of the medium between cuticle and the cover glass does not affect much (Supplementary Figure 1).

Second, we have tried sucrose solutions and halocarbon oil. The key feature of the mounting medium is not the refractive index. “Fly viability” is the key feature of the mounting medium for long-term intravital imaging. Sucrose and hydrophobicity oil could easily kill flies after release from the FlyVAB. They are very sticky and not easily wiped off by paper. As we pointed out in the main text, we also have tried glycerol and glue as the mounting medium. Flies were more readily released from H₂O and HA compared with glycerol and glue after intravital imaging.

H₂O is the best mounting medium for fly viability after imaging. We even don't need to clean fly abdomen after release. However, 2ul H₂O evaporates too easily to maintain sufficient imaging time. That's why we choose HA as the mounting medium.

Indeed, we tested different HA concentrations to find a balance between imaging quality and maintaining fly viability after FlyVAB. 10ug/ml HA solution is the best choice of mounting medium we know so far. It is possible that there may be better solutions that provide a similar survival rate and have better RI characteristics, but as far as our current imaging quality is good enough to clearly see what we want to see, we feel it's not necessary to continue exploring new mounting mediums.

2. mCherry labeling of ee cells is unreliable - It doesn't seem reliable to identify ee cells according to the level of expression of red fluorophore. I suggest using a different fluorophore for these important landmarks. If that is not an option, you could tentatively distinguish between the mCherry and tdTomato with spectral unmixing, if your microscope is capable, rather than relying on signal intensity. The safer alternative is to redesign your flies so that all the labels aren't red. Try miRFP703, for example.

A: Thank you for your professional suggestions.

We showed that, using a low 561 nm laser power (0.01% of the 561nm laser power by our scope) and a low digital gain (500V) to acquire mCherry signals, the ee cell marker Pros staining is colocalized with the *pros-QF>20XQUAS-6XmCherry* (Supplementary Figure 3b). Each Pros+ cell corresponds to a mCherry+ cell.

To visualize the tdTomato signals, we have to use 1.5% of the 561nm laser power and DG 650V to acquire the images. Considering that 150 times the laser excitation energy difference and 150V signal amplifying difference, there is no way to observe tdTomato signals under mCherry acquiring settings (LP 0.01%, DG 500V) (Supplementary Figure 5). Therefore, problem of “the level of expression of red fluorophore” does not exist in identifying ee cells by mCherry.

Nevertheless, the purpose of the *pros-QF>20XQUAS-6XmCherry* is not to identify ee cells. We used the pattern of mCherry as coordinates to re-identify the same intestinal region in a series of FlyVAB settings. We are satisfied with mCherry signals as coordinates (Fig. 5a).

We agree with reviewer that a stock like QUAS-miRFP703 would be great helpful for our research. There are limited QUAS-fluorophore stocks in Bloomington. We had tried to make a QUAS-YFP on the third chromosome. However, the plasmid we ordered from addgene #46162 pQUASp plasmid was not working in the intestinal epithelium. We communicated with Dr.Potter. We are using addgene #104880 pQUAST to make new QUAS stocks following Dr. Potter's suggestion.

3. Signaling in the abdomen – The authors conclude that damaged or aging abdominal tissues provide different signals to ISCs than young, health tissue does. They cite other work that suggests mechanisms where tissue physiology could affect Notch signaling, however, those signaling mechanisms are not addressed in these experiments. Perhaps the authors could design an experiment that explicitly compares aging and damaged tissue. Since you have optical access to the gut, maybe try photo-ablation or optogenetic manipulation of identified cells. The title of the paper mentions local physiology but the paper doesn't manipulate local physiology. That's an opportunity missed.

A: Thank you for your insightful suggestions. We saw this is an opportunity to manipulate stem cell behavior in real time. We have tried to induce real “local injury” under FlyVAB:

1) We tried burn cells of interest by laser ablation. However, the gut is an internal organ--there is an abdomen chitin cuticle (and gut muscle) blocking the laser energy to the epithelium. We failed killing enterocytes or ISCs directly by laser-ablation.

2)We ordered *UAS-miniSOG2* stock from Bloomington (BL#67609)[6]. We combined this *UAS-miniSOG2* with *esg-Gal4* or *Myo1A-Gal4* to test if ISCs or ECs could be ablated via photogenerated reactive oxygen species. Unfortunately, we couldn't induce cell ablation even after 10 mins illumination (regions of interest were rechecked multiple times after illumination).

We realized that our method supplied a good opportunity to understand “local injury” by intravital imaging. We believe it could be achieved in the near future using our FlyVAB method.

4. Figure issue – The figures are too heavily annotated to make interpretation possible. Decrease the thickness and brightness of outlines

A: We had tried our best to decrease the annotations.

Other issues

1. Your wording makes it sound like you invented QUAS.

A: Thank you for reminding us. We cited Dr. Potter's 2016 Genetics paper. We will add Dr. Potter and Dr. Luo's QF-QUAS original method paper [7-9] in the text.

2. Why did you do calcium imaging? Was it just to show that you could? How do the results compare to images acquired from gut explants or bellymount gut? If your results are similar, it would be another piece of evidence that your method maintains normal physiology. Also, is 0.5Hz fast enough to see actual calcium dynamics in the gut that aren't associated with cell death? I would also suggest jRGECO1b instead of GCaMP6s for imaging through the abdominal wall because of the decreased scattering at the longer wavelengths. Flies are available from Bloomington.

A: Thanks for your suggestion of the jRGECO1b stock. We ordered this stock in case GCaMP6s is not sensitive enough for research objectives in the future.

While the paper of bellymount didn't show calcium imaging data, comparing with gut explants data in a Nature paper from Dr. Jasper's group [10], they acquired images with 15 seconds intervals, lower than 0.067Hz (considering image acquiring time). Therefore, our 0.5Hz image acquiring improved the time resolution nearly by an order of magnitude.

There are two reasons that why our method acquiring images much faster than guts explants:

1. Gut explants shake in the culture medium due to muscle spontaneous contractions. Therefore, focal plane will be lost at different time points at Z-axis. Imaging acquiring for gut explants, for example, Jasper group recorded Z-stacks comprising 17 4- μm - spaced planes[10]. FlyVAB kept fly midgut relatively immobile, 4 6- μm -spaced planes were enough for time lapse movie acquiring.
2. Gut explants are in *ex vivo* conditions. Gut explants cannot deal with phototoxicity well. Midguts in FlyVAB could tolerate high frequency of confocal lasers as they are alive in the abdomen.

We also have tried fast image acquiring frequency (10 Hz for 30 second) using our Zeiss LSM800 microscope. We didn't find significant difference of calcium oscillation between 10Hz and 0.5Hz in such conditions. We concluded that 0.5Hz image acquiring is fast enough for ISC calcium oscillations.

Back to your first 3 questions and comments. Calcium imaging from gut explants showed that the fastest oscillating frequency was about 1 spike per minute, and the average oscillating frequency was about 1 spike every 2 minutes in control gut explants under normal physiology conditions (Figure 2b and Figure 3a in [10]). However, our calcium imaging showed a different calcium oscillation pattern in terms of frequency, amplitude, and duration comparing with the gut explants (Figure 3d'). We believe the

calcium signal from undissected gut is more likely to represent the real calcium signaling in normal physiology. We demonstrated that FlyVAB is applicable on short interval time lapse movies such as Ca^{2+} oscillation. This method could be used for anyone interested in imaging through the abdominal wall of living flies.

Figure 2b from Deng et.al. 2015 Nature

Figure 3d' from this manuscript

3. Many of the conclusions about cell fate are based on very low sample sizes ($n=1$). If you are going to make conclusions about the low impact of your method, you will need properly quantified data and appropriate statistical tests.

A: To make conclusions about the low impact of our method to the normal physiology, food consumption and defecation assay were performed in 10 independent groups separately. We carefully quantified the data, and results are shown as mean \pm SD. Plus, every number were plotted in the graph (Figure 2). The equal variance t-test was performed using GraphPad Prism 8.

Intravital tracking of quiescent ISCs and “stand-by” EBs in young and healthy flies were analyzed in 9 ISC-EB events ($n=9$). As a comparing, Bellymont method published in Elife 2020 analyzed 8 independent clones ($n=8$) (Fig 3 in [5]).

We showed one example ($n=1$) of ISC-EB-EC differentiation in Fig. 5. In this case, we demonstrated that the ee coordinates could help us to trace the EB to EC process.

4. Why didn't you cut the wings off of the control flies in the survival experiment? That's the only way to draw any conclusions about abdominal compression.

A: We understand your point, you were suggesting us that we should demonstrate the effect of abdominal compression to the survival rate. Therefore, we should compare with control flies without wings.

Since all the fly intestine studies were performed using flies with wings, it would be natural to argue that our wings off, abdominal compression flies had different life spans with control flies with wings. Even we demonstrated that there is no survival difference between abdominal compression flies with wings off flies, one can still argue our abdominal compression fly intestines couldn't represent fly intestines with wings.

Our data showed that there is no survival difference between abdominal compression flies and control flies with wings before 40 days old (Figure 2. d). We think this piece of data could help dispelling doubts.

Replies to Reviewer 3:

Reviewer #3 (Remarks to the Author):

In this manuscript, Guo and colleagues described a new method for intravital imaging in *Drosophila* and applied it to trace ISC-EB-EC lineages in midgut of young and old flies. The authors first designed a 3D-printed platform—"FlyVAB" and used hyaluronic acid as a mounting medium for periodic in vivo tracing over two weeks. Then by using distinctive patterns of ISCs and EEs as reference coordinates, they traced ISC-EB-EC lineage in the R4 region and found that in young and healthy flies, ISCs divide slowly and most EBs are in a "stand-by" status, while in aged flies with ISC hyperplasia, ISCs undergo rapid mitotic divisions and EBs are in a "ready-to-go" status for rapid differentiation into ECs.

The method developed in this study is an alternative to the previously developed "Bellymount" method, but with improved length of tracking time and easiness in operation. The finding of two different status of intestinal progenitors with this new method is also very interesting. Listed below a few minor points for authors to consider.

We really appreciated for the comments from the reviewer.

1. In the method part the authors described that they examined each image file at individual z-section and chose one representative" image of the labeled region, which came out of Fig.3-6 and some supplementary figures. But the authors need to make sure that the chosen images are truly "representative" for analyzing florescent intensity. What does it look like if acquiring the Z-stack images covering most of the thickness of the targeted cells and processing maximum or average projection of these images?

A: Thank you for your suggestions. The criterion for selecting representative images is that both of the ISC and EB are clear in the same focal plane. For example: the ISC-EB pair at 24h is Sharp-edged in Fig.5a. This image is the z5 focal plane of a series of 8 Z-stack images (Supplementary Figure 6a). We showed the z4 and z6 focal plane at 24h in Supplementary Figure 6b. ISC was off-plane in z4 and EB was off-plane in z6. We

have made this criterion clear in the method.

The reason we didn't show projection images of the Z-stack is that gut was moving when we acquiring the confocal images using a high-resolution setting. Therefore, the projection of Z-stack images covering most of the thickness was ghosted as below.

2. With live imaging, the authors observed that ISCs divide much slower than previously-observed and are close to quiescent. Is it possible that some operations applied to the animals during the experimental process, such as the compression applied to the fly abdomen, may have an impact on ISC behavior and thus cause ISC quiescence?

A: We think that the possibility of slowing down the division of ISCs caused by abdominal compression is very low. If abdominal compression could induce ISC quiescence, we shouldn't observe high frequency of ISC divisions in aged fly intestines by FlyVAB.

In addition, based on the observation in our lab and in the literature, the ISC division index, PH3+ cell number is always below 3 in young and healthy midguts. If PH3 could be visible in ~20mins of the M phase, 3 PH3+ cells/gut at any given dissection time means that there are at most: 24 hours/day X 60 mins/hour X (60mins/20 mins) X (3 PH3+/20mins)=216 PH3+/day. Considering there are more than 1000 ISCs per midgut, the majority of the ISCs should be quiescent in young and healthy flies. Since our intravital lineage tracing results revealed a slow ISC division rate in the intestines of young and healthy flies over a time course of 2-4 days, we think the slow ISC division rate observed in the intravital imaging is consistent with the no-compression-intestines' PH3+ immunostaining results.

3. Literally it is not obvious about the difference between "stand-by" and "ready-to-go", how about "differentiation-poised" and "differentiation-activated" instead? Also, can EBs with these two status be separated by molecular or morphological markers, such as the levels of Notch activity?

A: Thank you for your suggestion. We replaced "stand-by" and "ready-to-go" to "differentiation-poised" and "differentiation-activated".

These two statuses of EBs cannot be distinguished by different level of Notch activities. Similar Notch levels could be found in "differentiation-poised" (Fig. 4d) and "differentiation-activated" (Fig.6 and Supplementary Figure 7) EBs. We conclude that these two statuses of EBs are distinguished by the speed of Notch elevation: slowly rising of Notch activity represents the "differentiation-poised" and sharp rising of Notch activity represents the "differentiation-activated".

4. Figure 3A schematic: Similar to EC, EE cells are also derived from ISCs via a transient progenitor (EEP) stage.

A: We added EEP in this schematic cartoon.

5. The authors need to check carefully for spelling and formatting errors.

A: We have carefully improved the manuscript.

Reference:

1. Ohlstein, B. and A. Spradling, *Multipotent Drosophila intestinal stem cells specify daughter cell fates by differential notch signaling*. Science, 2007. **315**(5814): p. 988-92.
2. Ohlstein, B. and A. Spradling, *The adult Drosophila posterior midgut is maintained by pluripotent stem cells*. Nature, 2006. **439**(7075): p. 470-4.
3. Micchelli, C.A. and N. Perrimon, *Evidence that stem cells reside in the adult Drosophila midgut epithelium*. Nature, 2006. **439**(7075): p. 475-9.
4. Martin, J.L., et al., *Long-term live imaging of the Drosophila adult midgut reveals real-time dynamics of division, differentiation and loss*. Elife, 2018. **7**.
5. Koyama, L.A.J., et al., *Bellymount enables longitudinal, intravital imaging of abdominal organs and the gut microbiota in adult Drosophila*. PLoS Biol, 2020. **18**(1): p. e3000567.
6. Makhijani, K., et al., *Precision Optogenetic Tool for Selective Single- and Multiple-Cell Ablation in a Live Animal Model System*. Cell Chem Biol, 2017. **24**(1): p. 110-119.
7. Riabinina, O., et al., *Improved and expanded Q-system reagents for genetic manipulations*. Nat Methods, 2015. **12**(3): p. 219-22, 5 p following 222.
8. Potter, C.J. and L. Luo, *Using the Q system in Drosophila melanogaster*. Nat Protoc, 2011. **6**(8): p. 1105-20.

9. Potter, C.J., et al., *The Q system: a repressible binary system for transgene expression, lineage tracing, and mosaic analysis*. Cell, 2010. **141**(3): p. 536-48.
10. Deng, H., A.A. Gerencser, and H. Jasper, *Signal integration by Ca(2+) regulates intestinal stem-cell activity*. Nature, 2015. **528**(7581): p. 212-7.

Reviewers' comments:

Reviewer #2 (Remarks to the Author):

I understand that the authors are under some pressure to publish this work, and they worked long and hard on this project, and that many experiments that were not described went into the development of this technique. We can take it as given that all authors of all research papers are influenced by the same pressures. All of those factors, however, do not make a paper any more clear or worthy of publication. It is still unclear who the audience is or the point that this paper is trying to make.

Is this a "methods" paper? If so, it would be better presented as a technique for live-imaging in the fly abdomen with a diverse assortment of examples of great images. It has the advantage of being gentle enough to preserve the normal physiology of the specimen and precise enough to track nuclei in an intact fly. It could be presented that way without trying to include meaningful biological findings about stem cell biology.

Is this a "stem cell" paper? If so, there needs to be much more information about the different states of the ISCs and the physiology of the transition between EB and EC. The live-imaging technique could easily be an important method in that story, but the way it is presented, it isn't much of a stem cell story. Also, the stem cell story doesn't benefit from results that don't directly contribute to it, such as the calcium imaging. It's just confusing.

To me, this looks like a paper describing a live-imaging technique. Sell it that way: here is a great new technique and here are examples of the types of data that can be acquired by using it. Make this a strong methods paper rather than a weak stem cell paper.

Reviewer #3 (Remarks to the Author):

The authors have addressed my concerns satisfactorily. Language checking is still needed, for examples: there is an "a" missing from "China" in the author affiliation, the "esg-Gal" in the fourth line of Methods section should be "esg-Gal4", and "Drosophila" should be italic throughout the main text.

Reviewer #4 (Remarks to the Author):

Review of "An intravital imaging strategy revealing the dependence of enteroblast differentiation on the local physiology in the *Drosophila* intestine."

In this manuscript, Tang et al. details the development and use of "FlyVab." A device that is 3D printed, acts as a stage to live image *Drosophila melanogaster*, and allows an investigator to restrain an individual fly for live imaging.

Using this approach in combination with fluorescent reporter lines and tissue specific drivers of expression, the authors make observations to characterize enteroblast homeostasis in *Drosophila*. Through their results, they indicate that ISC-EB-EC cell dynamics is different in young, healthy *Drosophila* compared to older flies with age-associated hyperplasia. In young flies, enteroblast are in a "differentiation poised" state resulting in slow turnover, rare mitotic events, and no differentiation. Conversely, older flies are in a "differentiation activated" state and present increased turnover and differentiation.

The authors conclude that Notch signaling spikes in older flies and regulates ISC cell proliferation and differentiation. This conclusion is made by observing Notch signaling and EC cell formation with

fluorescent reporters in young and old flies. GFP fluorescence intensity was used to determine Notch signaling dynamics and different tissue specific drivers were used to identify undifferentiated ISC cells and differentiated EE and EC cells.

Overall, I like the creativity in making a new 3D printed device to improve live imaging in *Drosophila*. Files for 3D printing can be easily shared and other investigators can also use FlyVab. The illustrations in the manuscript are well made and make clear what is expected from the data. However, beyond this point, the manuscript appears rushed because of the grammatical and formatting errors (e.g., having two periods at the end of a sentence), and the paper lacks rigor to make some of the biological conclusions made in the paper. I list my concerns below.

1. Claiming that HA makes the abdomen clear.

From the supplementary data, it is obvious that HA performs almost the same as water. The authors do mention that it helps keep the fly in place and that it does not evaporate as fast as water. This is fine. But, the manuscript states that "the ventral abdomen cuticle becomes transparent due to the hyaluronic acid used as a mounting medium." This seems not to be the case.

2. Claiming that the "enteroblast differentiation process" is a result of the local physiology in the intestine.

While the authors did image older flies, and arguably older flies have different physiology, this is not enough to claim that physiology is different. In the older flies, no marker or assay was used to indicate how young and old physiology are different or whether any pathology is present. From this point of view, it looks like the differentiation event observed in older flies could have been normal physiology. A good experiment here would be to first show that older flies are experiencing age-related hyperplasia and that they have increased differentiation events when compared to young flies. Then, introduce stress in young flies and determine if differentiation increases. The paper cites that tissue aging or injury can lead to ISC proliferation and differentiation. It may not be possible to exactly recreate aging hyperplasia in young flies, but cell damage could be induced by something as simple as a thin needle poke, laser ablation, or a pharmacological reagent. With this experiment, the observation of young flies with altered physiology having increased differentiation would be more conclusive.

3. The use of QUAS-mCherry and saturated images.

The development of a QF/QUAS binary system expressing line is interesting and the use of the HACK method can be powerful in generating novel binary expression systems. However, this technology appears out of place in this manuscript. It doesn't appear to benefit the study. The system being used is a 20xQUAS-6xmCherry which is highly overexpressed. This can be beneficial in some applications. However, it seems this why very low laser power and exposure time was used to image EE cells only, and consequently resulting in over-exposed/saturated images when collecting data when EE, EB, and ISC cells are all present. I believe this is not the best experimental design. First, since fluorescence intensity ratios are being used to make biological conclusions, it is not good to have saturated pixels in the image. And secondly, the data suggests these drivers operate at different capacities. It would be more beneficial to compare similar drivers, rather than making conclusions on drivers that perform very differently.

4. The role of Notch signaling.

The conclusion that Notch signaling is part of the mechanism that regulates ISC proliferation and differentiation is not substantiated. The manuscript only reports two samples for observing NRE-GFP+ positive cells in older *Drosophila*. The quantification of Notch signaling is based on GFP intensity normalized to background, but this is calculated from saturated images. And finally, the experiments lack controls. For example, a good experiment here would be to use a Notch inhibitor in older flies and observe whether differentiation is also inhibited. Or, a similar strategy can be used with an over-expression of Notch and observing an increase in differentiation.

5. Additional Comments

Please check the use of genetic nomenclature. For example, I wasn't sure if you meant UAS-H2B-RFP or if this was a different line that has a constitutively active reporter. I have the same comment for NRE-GFP.

Figure 3, Panel B I don't see a difference between confocal and widefield images. Not sure how the higher resolution modality helps show more information.

It may be a better approach to photobleach a small area and use that as a coordinate reference rather than looking for the same pattern for multiple days.

6. Conclusion

I really like the creation of FlyVab. This can be a great tool for live imaging and makes imaging the *Drosophila* gut accessible to the community. However, in its current state, this manuscript is not ready for publication and needs additional work to make conclusions for the role of Notch signaling in ISC proliferation and differentiation.

We would like to express our sincere thanks to the reviewers for the constructive and positive comments.

Replies to Reviewer 2:

Reviewer #2 (Remarks to the Author):

I understand that the authors are under some pressure to publish this work, and they worked long and hard on this project, and that many experiments that were not described went into the development of this technique. We can take it as given that all authors of all research papers are influenced by the same pressures. All of those factors, however, do not make a paper any more clear or worthy of publication.

Dear reviewer #2,

Thanks for your understanding. I totally agree with you that pressure itself does not make paper clearer or worthier for publication.

It is still unclear who the audience is or the point that this paper is trying to make.

Is this a "methods" paper? If so, it would be better presented as a technique for live-imaging in the fly abdomen with a diverse assortment of examples of great images. It has the advantage of being gentle enough to preserve the normal physiology of the specimen and precise enough to track nuclei in an intact fly. It could be presented that way without trying to include meaningful biological findings about stem cell biology.

Is this a "stem cell" paper? If so, there needs to be much more information about the different states of the ISCs and the physiology of the transition between EB and EC. The live-imaging technique could easily be an important method in that story, but the way it is presented, it isn't much of a stem cell story. Also, the stem cell story doesn't benefit from results that don't directly contribute to it, such as the calcium imaging. It's just confusing.

To me, this looks like a paper describing a live-imaging technique. Sell it that way: here is a great new technique and here are examples of the types of data that can be acquired by using it. Make this a strong methods paper rather than a weak stem cell paper.

Answer: Thanks for your suggestions! To emphasize the technique and weaken the importance of the stem cell part, we modified many places in the parts of abstract, results and discussion. For example, we changed the last sentence of abstract

“Our results indicate that the enteroblast differentiation process could be from multiple hours to multiple days depending on the local physiology of the intestine” to “Our FlyVAB imaging strategy opens the door to long-time intravital imaging of intestinal epithelium” .

Although we would like to structurally change the manuscript to fulfil your suggestions, we found it is difficult to sell the FlyVAB method without the stem cell part.

The starting point of our research is to track the proliferation and differentiation of the same stem cell in the *Drosophila* intestinal epithelium for several days. To accomplish this goal, our method should include two parts: (1) a setup that could intravitaly image the intestinal epithelium; and (2) a tracking method that could faithfully identify the same intestinal stem cell and its progenies among multiple removal and squeezing operations.

Fig. 1 to Fig. 3 a-d, and Supplementary Figure 1 to Supplementary Figure 4 are the first part. We found a suitable media between fly cuticle and cover glass, making the cuticle of the fly abdomen pressed against the cover glass transparent. Using a 3D printing platform and a two-step process to close the lid, we could constantly visit the same region of the gut after releasing flies off from the FlyVAB.

To accomplish the second objective, we developed two methods to identify the same ISC among days: (1) Using the unique patterns of ISCs as reference coordinates, we identified 4 continuous time points of the same intestine region in stem cell quiescent intestines (Fig.4); and (2) In aged intestines that stem cells undergo quick divisions, we used the pattern of adult EEs as coordinates to re-identify the same intestinal region in a series of FlyVAB settings (Fig. 5 and 6, and Supplementary Figure 5-7).

In summary, if we don't present the findings in tracking the division and differentiation of intestinal stem cells, we feel that it is difficult to clearly show the advantages of our current FlyVAB methods. According to your suggestion, we have emphasized the technical part more in this manuscript, while weakening the meaning of stem cell division and differentiation. Hope to get your approval.

Replies to Reviewer 3:

Reviewer #3 (Remarks to the Author):

The authors have addressed my concerns satisfactorily. Language checking is still needed, for exmaples: there is an “a” missing from “China” in the author affiliation, the “esg-Gal” in the fourth line of Methods section should be “esg-Gal4”, and “Drosophila” should be italic throughout the main text.

Dear reviewer #3,

Sorry for our mistakes. We have checked the spelling again. Thanks for your

Careful reading.

Replies to Reviewer 4:

Reviewer #4 (Remarks to the Author):

Review of “An intravital imaging strategy revealing the dependence of enteroblast differentiation on the local physiology in the Drosophila intestine.”

In this manuscript, Tang et al. details the development and use of “FlyVab.” A device that is 3D printed, acts as a stage to live image Drosophila melanogaster, and allows an investigator to restrain an individual fly for live imaging.

Using this approach in combination with fluorescent reporter lines and tissue specific drivers of expression, the authors make observations to characterize enteroblast homeostasis in Drosophila. Through their results, they indicate that ISC-EB-EC cell dynamics is different in young, healthy Drosophila compared to older flies with age-associated hyperplasia. In young flies, enteroblast are in a “differentiation poised” state resulting in slow turnover, rare mitotic events, and no differentiation. Conversely, older flies are in a “differentiation activated” state and present increased turnover and differentiation.

The authors conclude that Notch signaling spikes in older flies and regulates ISC cell proliferation and differentiation. This conclusion is made by observing Notch signaling and EC cell formation with fluorescent reporters in young and old flies. GFP fluorescence intensity was used to determine Notch signaling dynamics and different tissue specific drivers were used to identify undifferentiated ISC cells and differentiated EE and EC cells.

Overall, I like the creativity in making a new 3D printed device to improve live imaging in Drosophila. Files for 3D printing can be easily shared and other investigators can also use FlyVab. The illustrations in the manuscript are well made and make clear what is expected from the data. However, beyond this point, the manuscript appears rushed because of the grammatical and formatting errors (e.g., having two periods at the end of a sentence), and the paper lacks rigor to make some of the biological conclusions made in the paper. I list my concerns below.

We really appreciated for the comments from the reviewer. Sorry for the grammatical and formatting errors, we have checked the grammar and spelling again.

1. Claiming that HA makes the abdomen clear.

From the supplementary data, it is obvious that HA performs almost the same as

water. The authors do mention that it helps keep the fly in place and that it does not evaporate as fast as water. This is fine. But, the manuscript states that “the ventral abdomen cuticle becomes transparent due to the hyaluronic acid used as a mounting medium.” This seems not to be the case.

Answer: After trying different medium, we realized that when the fly abdomen is pressed against the cover glass, the gap between the abdomen cuticle and the cover glass produces light scattering, making the light unable to pass through the cuticle for imaging. H₂O or 10% HA solution we used fills these gaps, thereby making the cuticle of the fly abdomen pressed against the cover glass transparent (Figure 1C).

We have tried sucrose solutions and halocarbon oil. However, “Fly viability” is the key feature of the mounting medium for long-term intravital imaging. Sucrose and hydrophobicity oil could easily kill flies after releasement from the FlyVAB. They are very sticky and not easily wiped off by paper. We also have tried glycerol and glue as the mounting medium. Flies were more readily released from H₂O and HA compared with glycerol and glue after intravital imaging.

H₂O is the best mounting medium for fly viability after imaging. We even don't need to clean fly abdomen after releasement. However, 2ul H₂O evaporates too easily to maintain sufficient imaging time. That's why we choose HA as the mounting medium.

We also tested different HA concentrations to find a balance between imaging quality and maintaining fly viability after FlyVAB. 10ug/ml HA solution is the best choice of mounting medium we know so far.

2. Claiming that the “enteroblast differentiation process” is a result of the local physiology in the intestine. While the authors did image older flies, and arguably older flies have different physiology, this is not enough to claim that physiology is different. In the older flies, no marker or assay was used to indicate how young and old physiology are different or whether any pathology is present. From this point of view, it looks like the differentiation event observed in older flies could have been normal physiology. A good experiment here would be to first show that older flies are experiencing age-related hyperplasia and that they have increased differentiation events when compared to young flies. Then, introduce stress in young flies and determine if differentiation increases. The paper cites that tissue aging or injury can lead to ISC proliferation and differentiation. It may not be possible to exactly recreate aging hyperplasia in young flies, but cell damage could be induced by something as simple as a thin needle poke, laser ablation, or a pharmacological reagent. With this experiment, the observation of young flies with altered physiology having increased differentiation would be more conclusive.

Answer: Thank you for your insightful suggestions. We could induce tissue damage and intestinal hyperplasia by bleomycin treatment^{1,2}. Therefore, to visualize the

physiology change in aged (14 days old) *Drosophila* intestines, we crossed *upd3-lacZ*, an EC damage and turnover indicator^{3,4}, and *vn-lacZ*, a midgut hyperplastic marker in muscle cells³ with *esg-Gal4 10×UAS-myr:tdTomato*. 14 days old intestines had more *upd3-lacZ* positive ECs than young intestines and much less *upd3-lacZ* positive ECs than bleomycin injured intestines (Supplementary Fig. 7a). *vn-lacZ* in muscle cells was significantly increased in aged intestines than in young intestines, and *vn-lacZ* had the strongest signal intensity in injured intestinal muscle cells (Supplementary Fig. 7a and 7c). By counting the ISC mitotic division marker Phospho-Histone H3 (PH3)⁴ in *esg-Gal4 10×UAS-myr:tdTomato, NRE-GFP; pros-QF2 20×QUAS-6×mcherry* flies, we found the number of dividing ISCs was significantly increased in aged intestines than in young ones, and the highest PH3 positive ISC number was found in bleomycin injured intestines (Supplementary Fig. 7b and 7d). Taken together, our data suggested that our 14 days old fly intestines were in a mild hyperplasia state compared with young fly intestines and bleomycin injured intestines.

Supplementary Figure 7 a-d. 14 days old intestines were in a mild hyperplastic status

a Representative images of *upd3-lacZ*, *vein-lacZ* staining in young (7 days), aged (14 days) and injured (1 day after bleomycin treatment) flies. Genotype: *esg-Gal4 10 × UAS-myr:tdTomato/upd3-lacZ* or *vn-lacZ*. **b** Representative images of PH3 staining in young, aged and injured flies. White arrowhead: PH3+ cell. Genotype: *esg-Gal4 10×UAS-myr:tdTomato, NRE-GFP; pros-QF2 20×QUAS-6×mcherry*. **c** Quantification of the *vein-lacZ* intensity in **a**. Results are shown as mean \pm SD. **d** Quantification of PH3+ cells per midgut in young, aged and injured flies. Results are shown as mean \pm SD. Scale bars are 20 μ m in **a** and **b**.

Adult intestines that over 20 days old have a much more serious gut injury phenotype and gut hyperplasia^{5,6} than 14 days old intestines. Besides bleomycin,

we could also induce tissue damage and hyperplasia in young fly intestines using paraquat¹, DSS¹, and gram-negative bacterium *Erwinia carotovora*⁷ (pictures are shown below). We have tried different aged fly intestines and tested different injury methods in young flies to observe daughter cell differentiation and ISC division by FlyVAB. However, there were over crowded *esg>GFP* progenitors in >20 days old fly intestines or injured ones. Those progenitors quickly divided, continuously changed their positions and geometry over time. Under those conditions, even EE cells were also quickly generated and turnover, making the long-time intravital tracking impossible. Therefore, we selected the 14 days old fly intestines as a compromise to track the daughter cell differentiation and ISC division using FlyVAB.

We also have tried to induce real “local injury” under FlyVAB:

1) We tried burn cells of interest by laser ablation. However, the gut is an internal organ---there is an abdomen chitin cuticle (and gut muscle) blocking the laser energy to the epithelium. We failed killing enterocytes or ISCs directly by laser-ablation.

2) We ordered *UAS-miniSOG2* stock from Bloomington (BL#67609)⁸. We combined this *UAS-miniSOG2* with *esg-Gal4* or *Myo1A-Gal4* to test if ISCs or ECs could be ablated via photogenerated reactive oxygen species. Unfortunately, we couldn’ t induce cell ablation even after 10 mins irradiation (regions of interest were rechecked multiple times after irradiation).

We realized that our method supplied a good opportunity to understand “local injury” by intravital imaging. We believe it could be achieved in the near future using our FlyVAB method.

3. The use of QUAS-mCherry and saturated images.

The development of a QF/QUAS binary system expressing line is interesting and the use of the HACK method can be powerful in generating novel binary expression systems. However, this technology appears out of place in this manuscript. It doesn’ t appear to benefit the study. The system being used is a 20xQUAS-6xmCherry which is highly overexpressed. This can be beneficial in some applications.

However, it seems this why very low laser power and exposure time was used to image EE cells only, and consequently resulting in over-exposed/saturated images when collecting data when EE, EB, and ISC cells are all present. I believe this is not the best experimental design. First, since fluorescence intensity ratios are being used to make biological conclusions, it is not good to have saturated pixels in the image. And secondly, the data suggests these drivers operate at different capacities. It would be more beneficial to compare similar drivers, rather than making conclusions on drivers that perform very differently.

Answer: We have tried our best to invent a method to track the same region of the intestine at different time points over 10 days. Guts are constantly moving in the abdomen; plus, stem cells and their daughters are not static in the epithelium: stem cells are dividing, daughter cells are growing in size and moving in distance. In young fly intestines, as ISCs are relatively quiescent, we could track the same region of an intestine by a unique geometry pattern of different ISCs (Fig.4). But in 14 days old intestines, since ISCs divides faster and faster, the geometry patterns of ISCs were quickly lost in aged guts.

Since EE cells are relatively stable in young and aged intestines (but not in too old guts), after long time thinking and testing, we decided to label the EE cells as coordinates to trace the same region of an intestine. To achieve this goal, we had to construct a fly stock with three different markers: ISC and EB marker (*esg>tdTomato+* cells), Notch signaling reporter (*NRE-GFP*), and an independent EE cell marker. We spent 6 months to generate a *pros-QF* stock using the HACK method. Ideally, *pros-QF* driving a far-red fluorophore as coordinates would be elegant (to avoid “*drivers operate at different capacities*”). However, there are limited QUAS-fluorophore stocks in Bloomington fly stock center. We had tried to make a QUAS-YFP by ourselves. However, the plasmid we ordered from addgene #46162 pQUASp plasmid was not working in the intestinal epithelium. We communicated with Dr.Potter, who donated this plasmid. We are using addgene #104880 pQUAST to make new QUAS stocks following Dr. Potter’ s suggestion.

The remaining options for QF were driving either a green-fluorophore or a red-fluorophore to label the EE cells. Because we don’ t want to mess up with the NRE-GFP channel (“*fluorescence intensity ratios are being used to make biological conclusions*”), we decided to express a super bright red fluorophore by *pros-QF* to distinguish EE cells from *esg>tdTomato* labeled ISC and EBs.

First, a low laser power and a low digital gain allowed the visualization of mCherry labelled EEs, but not tdTomato labelled cells. Then, a high laser power and high digital gain enable the visualization of the ISC-EB pairs, plus the overexposed EEs. We used the pattern of mCherry as coordinates to re-identify the same intestinal region in a series of FlyVAB settings. This is why we finally selected the *20xQUAS-6xmCherry* as the QF driving fluorophore to achieve our goal.

Back to your first concern: “*since fluorescence intensity ratios are being used to make biological conclusions, it is not good to have saturated pixels in the image.*”

As described above, the saturated pixels were obtained under the 561nm laser, while the NRE-GFP fluorescence signal was obtained using the 488nm laser. They were captured separately. We concluded that the GFP channel intensity would not be affected by the saturated pixels.

4. The role of Notch signaling.

The conclusion that Notch signaling is part of the mechanism that regulates ISC proliferation and differentiation is not substantiated. The manuscript only reports two samples for observing NRE-GFP+ positive cells in older Drosophila. The quantification of Notch signaling is based on GFP intensity normalized to background, but this is calculated from saturated images. And finally, the experiments lack controls. For example, a good experiment here would be to use a Notch inhibitor in older flies and observe whether differentiation is also inhibited. Or, a similar strategy can be used with an over-expression of Notch and observing an increase in differentiation.

Answer: Thank you for your insightful suggestions. The Notch signaling is well documented in regulating ISC proliferation and differentiation^{4,9-13}. In the initial literatures, Micchelli et. al. reported that knockdown of Notch using Notch RNAi caused ISC over-proliferation and blocking the differentiation of ECs (d)¹³. Over-expression of a constitutive active form of Notch receptor caused ISCs differentiating into ECs (f)¹³. Similar results were obtained in the back to back paper¹². Just like you suggested, Ohlstein et. al. used a Notch inhibitor (DAPT) to feed the flies, observed that ISCs and their daughters differentiated into Tachykinin positive enteroendocrine cells (c). Notch null mutant clone showed over-proliferation and differentiation defects (e)¹².

Micchelli et. al. 2006
Nature Figure 4

Ohlstein et. al. 2006
Nature Figure 4

In our lab conditions, we tested the knockdown of Notch and ectopic activation

of Notch in 14 days old flies (data is shown below). Consistent with the literature, knockdown of Notch signaling resulted in ISC over-proliferation and differentiation to ee cells, while ectopic expression of a constitutive active form of Notch (Notch^{intra}) resulted in the EC differentiation.

Although Notch signaling plays such an important role in intestinal stem cell proliferation and differentiation, how Notch signaling is activated in daughter cells is still obscure under normal/aging/injury physiology conditions. The Notch signaling was initially thought to be activated in the daughter of stem cells within a few minutes to promote the differentiation of EB into EC cells¹⁴⁻¹⁶. Through a cut-window in the abdomen of flies, Lucy O'brien's group found that activation of Notch took several hours¹⁷. Our non-invasive intravital imaging found that the activation of Notch varied from hours to dozens of hours depending on the physiology of the midgut. There are few examples in the stem cell field discussing the activation speed of Notch. People know that Notch affects stem cell differentiation, but our data suggested that the speed of activation matters. Furthermore, our imaging data is the first time observed that Notch signaling in the ISC daughter cells could be declined after the separation of ISC and EB (Fig.4 c). We believe the biology we found here is interesting for many stem cell and Notch signaling researchers.

5. Additional Comments

Please check the use of genetic nomenclature. For example, I wasn't sure if you meant UAS-H2B-RFP or if this was a different line that has a constitutively active reporter. I have the same comment for NRE-GFP.

Answer: Dear reviewer, thank you for your reminding. The *H2B-RFP* means *ubi-H2B:RFP*, which is a ubiquitously expressed H2B:RFP fusion protein. To make it clear, we have changed the nomenclature in the main text and method.

The *NRE-GFP* is correct. "*NRE*" is a genetic nomenclature of Notch Responds Element¹⁸. *NRE-GFP* means: three copies of the *grh* protein binding element *Gbe* and two *Su(H)* binding sites from the *E(sp1)* gene are fused upstream of a minimal

Hsp70Bb promoter driving GFP expression¹⁸. This *NRE* promoter is widely used in *Drosophila* ISC research field to monitor the Notch activation^{4,10,11,13,19,20}.

Figure 3, Panel B I don't see a difference between confocal and widefield images. Not sure how the higher resolution modality helps show more information.

Answer: Dear reviewer, the “widefield” in the Fig.3 b means images captured using “a widefield fluorescence microscope” (line 172). We compared the feasibility of imaging using the same FlyVAB setup under a confocal or a widefield/basic fluorescence microscope. We would like to emphasize that the method we developed is not dependent on confocal microscope. To make it clear, we changed the labels to “confocal microscope” and “widefield microscope” in Fig.3 b.

It may be a better approach to photobleach a small area and use that as a coordinate reference rather than looking for the same pattern for multiple days.

Answer: Thank you for your advice. We have tried to use a small photobleach area as a coordinate reference. However, fluorescence signals were recovered several hours after photo bleaching, making it impossible to track the same region for multiple days.

6. Conclusion

I really like the creation of FlyVab. This can be a great tool for live imaging and makes imaging the Drosophila gut accessible to the community. However, in its current state, this manuscript is not ready for publication and needs additional work to make conclusions for the role of Notch signaling in ISC proliferation and differentiation.

Answer: Thank you very much for your appreciation! And also, thanks for your inspiring suggestions.

References

- 1 Amcheslavsky, A., Jiang, J. & Ip, Y. T. Tissue damage-induced intestinal stem cell division in *Drosophila*. *Cell stem cell* **4**, 49-61, doi:10.1016/j.stem.2008.10.016 (2009).
- 2 Guo, Z., Driver, I. & Ohlstein, B. Injury-induced BMP signaling negatively regulates *Drosophila* midgut homeostasis. *The Journal of cell biology* **201**, 945-961, doi:10.1083/jcb.201302049 (2013).
- 3 Jiang, H., Grenley, M. O., Bravo, M.-J., Blumhagen, R. Z. & Edgar, B. A. EGFR/Ras/MAPK signaling mediates adult midgut epithelial homeostasis and regeneration in *Drosophila*. *Cell stem cell* **8**, 84-95, doi:10.1016/j.stem.2010.11.026 (2011).

- 4 Jiang, H. *et al.* Cytokine/Jak/Stat signaling mediates regeneration and homeostasis in the Drosophila midgut. *Cell* **137**, 1343–1355, doi:10.1016/j.cell.2009.05.014 (2009).
- 5 Biteau, B., Hochmuth, C. E. & Jasper, H. JNK activity in somatic stem cells causes loss of tissue homeostasis in the aging Drosophila gut. *Cell Stem Cell* **3**, 442–455, doi:10.1016/j.stem.2008.07.024 (2008).
- 6 Rodriguez-Fernandez, I. A., Tauc, H. M. & Jasper, H. Hallmarks of aging Drosophila intestinal stem cells. *Mechanisms of Ageing and Development* **190**, 111285, doi:10.1016/j.mad.2020.111285 (2020).
- 7 Buchon, N., Broderick, N. A., Kuraishi, T. & Lemaitre, B. Drosophila EGFR pathway coordinates stem cell proliferation and gut remodeling following infection. *BMC Biology* **8**, 152, doi:10.1186/1741-7007-8-152 (2010).
- 8 Makhijani, K. *et al.* Precision Optogenetic Tool for Selective Single- and Multiple-Cell Ablation in a Live Animal Model System. *Cell chemical biology* **24**, 110–119, doi:10.1016/j.chembiol.2016.12.010 (2017).
- 9 Patel, P. H., Dutta, D. & Edgar, B. A. Niche appropriation by Drosophila intestinal stem cell tumours. *Nature Cell Biology* **17**, 1182–1192, doi:10.1038/ncb3214 (2015).
- 10 Guo, Z. & Ohlstein, B. Bidirectional Notch signaling regulates Drosophila intestinal stem cell multipotency. *Science* **350**, doi:10.1126/science.aab0988 (2015).
- 11 Ohlstein, B. & Spradling, A. Multipotent Drosophila intestinal stem cells specify daughter cell fates by differential notch signaling. *Science (New York, N. Y.)* **315**, 988–992, doi:10.1126/science.1136606 (2007).
- 12 Ohlstein, B. & Spradling, A. The adult Drosophila posterior midgut is maintained by pluripotent stem cells. *Nature* **439**, 470–474, doi:10.1038/nature04333 (2006).
- 13 Micchelli, C. A. & Perrimon, N. Evidence that stem cells reside in the adult Drosophila midgut epithelium. *Nature* **439**, 475–479, doi:10.1038/nature04371 (2006).
- 14 Ohlstein, B. & Spradling, A. Multipotent Drosophila intestinal stem cells specify daughter cell fates by differential notch signaling. *Science* **315**, 988–992, doi:10.1126/science.1136606 (2007).
- 15 Ohlstein, B. & Spradling, A. The adult Drosophila posterior midgut is maintained by pluripotent stem cells. *Nature* **439**, 470–474, doi:10.1038/nature04333 (2006).
- 16 Micchelli, C. A. & Perrimon, N. Evidence that stem cells reside in the adult Drosophila midgut epithelium. *Nature* **439**, 475–479, doi:10.1038/nature04371 (2006).
- 17 Martin, J. L. *et al.* Long-term live imaging of the Drosophila adult midgut reveals real-time dynamics of division, differentiation and loss. *eLife* **7**, doi:10.7554/eLife.36248 (2018).
- 18 Furriols, M. & Bray, S. A model Notch response element detects Suppressor of Hairless-dependent molecular switch. *Current biology : CB* **11**, 60–64,

doi:10.1016/s0960-9822(00)00044-0 (2001).

- 19 Martin, J. L. *et al.* Long-term live imaging of the *Drosophila* adult midgut reveals real-time dynamics of division, differentiation and loss. *eLife* **7**, e36248, doi:10.7554/eLife.36248 (2018).
- 20 Liang, J., Balachandra, S., Ngo, S. & O' Brien, L. E. Feedback regulation of steady-state epithelial turnover and organ size. *Nature* **548**, 588-591, doi:10.1038/nature23678 (2017).